



# Historical record of the effects of anthropogenic pollution on benthic foraminifera over the last 110 years in Gamak Bay, South Korea

Da Un Jeong[1], Yeon Gyu Lee[1], Yong Wan Kim[2], Jung Jun Park[3], Jung Sick Lee[4],

[1]Faculty of Marine Technology, Chonnam National University, Yeosu, 59626, Republic of Korea
[2]Center for Research Facilities, Chonnam National University, Yeosu, 59626, Republic of Korea
[3]South Sea Fisheries Research Institute, NIFS, Yeosu, 59780, Republic of Korea
[4]Faculty of Aqualife Medicine, Chonnam National University, Yeosu, 59626, Republic of Korea

*Correspondence to*: Yeon Gyu Lee (lyg6342@jnu.ac.kr)

**Abstract.** This study investigated the historical record of the effects that anthropogenic pollution has had on benthic
foraminifera over the last 110 years in the semi-closed Gamak Bay. The evidence consisted of geochemical data including
$^{210}$Pb concentrations and benthic foraminiferal assemblages acquired from core sediments (western, eastern and northwestern
areas). Various records of benthic foraminiferal assemblage in the northwestern area were suitable as the standard for variation
regarding pollution history. In the period between 1906 and 1964 (the pre-urbanization period), Gamak Bay was composed of
*Ammonia beccarii-Elphidium advenum-Elphidium clavatum* assemblage, except for the northwestern area with *A. beccarii-*
*Buccella frigida-E. advenum* assemblage, and may have remained mostly unpolluted. Although the northwestern area did not
show a difference in the species composition of the benthic foraminifera, it may be polluted to some degree due to stagnant
sewage supplied from a small village that had formed before city construction in the hinterland, as shown from the species
diversity of 1.37, with a total number of benthic foraminifera (TNBF) of 704 individual and total organic carbon/total sulfur
(C/S) of 2.63.

The benthic foraminiferal assemblage of the northernmost area between 1965 and 1987 (the urbanization period) rapidly varied
from *E. somaense-A. beccarii-B. frigida* assemblage, through *A. beccarii-B. frigida-E. advenum, B. frigida-A. beccarii-E.
subarcticum, T. hadai-E. subarcticum-B. frigida*, to *A. beccarii-E. subarcticum-T. hadai* assemblage with a diversity of 1.8,
TNBF of 244, C/S of 2.05 on average. During this period, it was characterized by an increase in abundance frequency in *T.
hadai*, and *E. subarcticum*, which are known as bioindicators of eutrophication and organic pollution, respectively, and rapid
variation of benthic foraminiferal assemblage. These may have been caused by an increase in the influx of sewage from Yeosu
City, which was constructed at the hinterland of the northernmost area in Gamak Bay, as shown from the sedimentation rate
of 1.0 cm/y. Pollution during the urbanization period may have been restricted to the northwestern area, and it did not diffuse
to the surrounding area.

The *E. subarctum*, *A. beccarii-E. subarcticum-T. hadai* and *E. subarcticum* assemblages with diversity of 1.35, TNBF of 562,
C/S of 2.33 were sequentially distributed in the northwestern area from 1988 to 2014 (the aquaculture period), and this is
characterized by the high abundance frequency of E. subarcticum of 51% and high sedimentation rate of 1.75 cm/y caused by



biodeposits discharged from mussel farming (Mytilus galloprovincialis) since the 1980s. The organic pollution materials originated from and deposited by biodeposits may contribute to the continuous deterioration and variation in the benthic ecological environment by means of "pollution storage". During this period, benthic foraminiferal assemblages in the northwestern area are correlated with the *E. subarcticum-A. beccarii* assemblage of the eastern area where oyster farming has taken place, and it is composed of *E. subarcticum* (35.4%) and *A. beccarii* (15.5%) with a TNBF of 1787 individuals, species diversity of 2.18, C/S of 4.8 and a sedimentation rate of 0.95 cm/year. It is clear that the northern area seriously progressed in pollution compared to the eastern area, although the species composition is somewhat similar between the two areas. It may be caused by an overabundance and excessive deposition of the organic matter through an over-supply from mussel farming as well as the oval-shaped bottom physiography and very slow current speed. During the transition from the pre-urbanization to urbanization period, and aquaculture period in the northwestern area, the processes of variation in the benthic foraminiferal assemblages may represent the transition from oxic to anoxic environmental conditions. The western area with *A. beccarii-E. advenum-E. clavatum* assemblage, however, was unpolluted over the last 110 years. These differences in the degree of pollution and benthic foraminiferal assemblages between the areas in Gamak Bay may be caused by the physiography and current movements of the bay.

# 1 Introduction

Over the last few centuries, coastal areas have experienced a dramatic degradation in environmental quality as a result of anthropogenic activity, and this has led to a considerable reduction in marine biodiversity. The anthropogenic impact on coastal marine ecosystems has multiple origins, including the introduction of urban sewage, outflows from industrial and agricultural activities including fisheries, and other environmental problems including eutrophication, oxygen deficiency, chemical pollution, and physical disturbance (Barras et al., 2014; Yasuhara et al., 2012).

Sediments are an essential, integral, and dynamic part of an aquatic environment and can act not only as a sink for various environmental chemicals, but also as a potential long-term secondary source of pollutants (Ridgway and Shimmield, 2002). The geochemical study of vertical sediment cores has been extensively used to reconstruct environmental transformation of different coastal areas all over the world (Cearreta et al., 2002; Di Gregorio et al., 2007). In the sedimentary record, interactions between meiofaunal and geochemical elements have made it possible to distinguish between unimpacted, pre-industrial intervals and sediments deposited in industrial periods. (Francescangeli et al., 2016).

Benthic foraminifera have been commonly used in reconstructions of the environmental changes over the past several centuries, including changes caused by human activity (Scott et al., 2005; Tsujimoto et al., 2008; Irabien et al., 2008; Dolven et al., 2013; Romano et al., 2016; Francescangeli et al., 2016). Benthic foraminifera can become fossilized and can act as reliable indicators of environmental change over historical and geological time scales (Gooday et al., 2009). The fossil remains of benthic foraminifera can provide information of the long-term environmental and biological changes, whether natural or human-



induced (Alve et al., 2009). Foraminifera offer considerable advantages over other groups of benthic organisms because their hard shells (called "tests") often persist as a record in the sediment and their small size makes them abundant even in small-volume samples. Thus they can be considered a reliable data source for statistical purposes. These characteristics make foraminifera suitable indicators for environmental studies of sediment cores and offer the opportunity to study temporal

changes in the ecological conditions (Romano et al., 2016). Moreover, since environmental quality must be assessed through a comparison with reference conditions, sediment cores offer the advantage of including ancient deposits reflecting conditions from before human impact in lieu of searching pristine areas with the same characteristics as the targeted study area (Alve et al., 2009).

The coastal zones of Korea began to undergo urbanization in earnest in the 1910s with increased fossil fuel (especially coal)

consumption (Jeong et al., 2006; Lim et al., 2012, 2013). Gamak Bay, which is located at the center of the southern coastal area of Korea, is known to experience problems in its coastal environments, including eutrophication, hypoxia, and red tide events caused by an influx of anthropogenic pollutants, particularly organic matter (Lee et al., 2009; Lee et al., 2012, 2016; Seo et al., 2012). Historically, this relatively small bay has been subjected to a combination of environmental pressures that mainly arise from aquaculture and industrial activities, and it is therefore a good case study of historical variations in

anthropogenic impacts. The purpose of this study is to investigate historical anthropogenic pollution as recorded by benthic foraminifera over the last 110 years in Gamak Bay through an interpretation of geochemical data, including $^{210}$Pb concentrations and benthic foraminiferal assemblage data acquired from core sediments.

## 2 Study area

Gamak Bay is located at the center of the South Sea coast of Korea. It is an oval-shaped, semi-enclosed bay surrounded by

Yeosu and Dolsan islands (Figure 1). It has an area of 148 km$^2$, approximately 15 km in length and 9 km in width (Lee et al., 1995). The bedrock beneath Gamak Bay is composed mainly of Cretaceous alkali-feldspar granite, andesite and andesitic tuff, a volcanic rock of intermediate composition with an aphanitic texture (KIGAM, 2002). The surface sediments of Gamak Bay consist mainly of fine-grained silt and clay facies, although coarse-grained sediments are predominant at the mouth of the bay (Lee et al., 1995).

The average water depth of the bay is 9 m, with relatively shallow water (<5 m depth) in the center increasing to ~30 m at the mouth of the bay. The tide is semidiurnal, and the tidal fluctuation is quite large, with minimum and maximum tidal amplitudes of ~1 m and ~4 m during the neap and spring tides, respectively. The bay has two channels, one to the east and one to the south, but the southern channel is responsible for approximately 80% of all seawater exchange in the bay (Lee and Chang, 1982). Tidal waves enter or exit almost simultaneously via these two channels with relative proportions of exchange (Lee et

al., 2009). The sea water of the bay can be divided into three water masses: (1) water in the northwestern area with a current that generally flows counterclockwise with water that is quite stagnant due to the bottom topography, residual bottom currents





with mean velocity of about 2 cm·s$^{-1}$ (Lee, 1992), and oxygen deficiency in the bottom water during the summer (Lee et al, 2016); (2) Yeosu Harbor water in the northeastern area with a lower salinity due to the influence of small streams and ditches; and (3) water in the center and near the mouth of the bay with a generally clockwise current (Lee and Cho, 1990) and residual bottom currents with mean velocity of about 4 cm·s$^{-1}$ (Lee et al., 2016).

The bay is home to many commercially important marine organisms, including oysters, mussels, ark clams, rockfish, sea bream, and flounder. Mussels and oysters are cultivated mostly using suspended longline systems and have been farmed in the northwestern and eastern areas of Gamak Bay, respectively, since the late 1970s, which makes this area important to the fishing industry. This bay was designated by the Korean government (Ministry of Land Transport and Maritime Affairs, MLTM) as an environmental conservation area in February of 2000 based on its ecological importance as a habitat for fish and shellfish

(Kim, 2003). However, increased human activities with an expansion of nearby urban areas since the 1970s, as well as the development of various aquaculture industries including oyster and mussel farming in recent decades, have caused a gradual increase in the influx of anthropogenic pollutants, particularly organic matter. These pollutants have caused environmental problems at the coast, including eutrophication, hypoxia, and red tide events, especially during the summer (Lee et al., 2009; Seo et al., 2012; Lee et al., 2012, 2016). These phenomena have occurred continuously, even though a sewage treatment plant

began operation in 2004 to restrain the influx of sewage from urban areas (Kim et al., 2006; Lee and Moon, 2006; Lee et al., 2009). Much of the current pollution in Gamak Bay originates from aquaculture activities and polluted surface sediments (Lee et al., 2016).

## 3 Materials and methods

### 3.1 Sediment sampling

A core sediment sampling was conducted in August 2015 at three stations in representative localities of the western (St. 11257), eastern (St. 10863) and northwestern (St. 11285) areas of Gamak Bay (Figure 1). These three sediment cores, with lengths of 40 cm, 44 cm and 60 cm, respectively, were extracted using a gravity corer of 76 mm in diameter and were analyzed in 2-cm intervals to examine grain size, trace metals, organic matter (OM), total organic carbon (TOC), total nitrogen (TN), total sulfur (TS), pH, $^{210}$Pb radioactivity, and benthic foraminifera.

### 3.2 Grain size composition and geochemical analysis

Prior to a grain size analysis, the organic material and carbonates were eliminated from the sample by sequentially adding 10% hydrogen peroxide (H$_2$O$_2$) and 0.1 N hydrochloric acid (HCl). Subsequently, the samples were subjected to a Sedigraph 5100 automatic particle size analyzer to measure the fine fraction (<63 μm) and sieve analysis for the coarse fraction. The weights of the coarse and fine samples were recorded as weight percentages for each section (Folk, 1968).



For the trace metal analysis, the samples were freeze-dried and ground to a fine powder with an agate mortar. A 0.25-g portion of each sample was measured into a Teflon decomposition container, into which 6 mL of $HNO_3$ (65%), 1 mL of $HClO_4$ (65%), and 1 mL of $H_2O_2$ (30%) were added. The mixture was processed with the decomposition sequence of a microwave decomposition system (ETHOS TC, Milestone, Italy). After cooling, the mixture was diluted to 50 mL with 0.1% nitric acid.

These samples were used for inductively coupled plasma optical mass spectrometry (ICP-MS, NexION®300X). Only the concentrations of Al, Fe, Mn, Zn, Cr, Ni, Cu, Co, As, Cd, Pb, and Hg were considered. The detection limits for each of these metals were: Al, 0.005 µg kg$^{-1}$; Fe, 0.0003 µg kg$^{-1}$; Mn, 0.00007 µg kg$^{-1}$; Zn, 0.0003 µg kg$^{-1}$; Cr, 0.0002 µg kg$^{-1}$; Ni, 0.0004 µg kg$^{-1}$; Cu, 0.0002 µg kg$^{-1}$; Co, 0.0009 µg kg$^{-1}$; As, 0.0006 µg kg$^{-1}$; Cd, 0.00009 µg kg$^{-1}$; Pb, 0.00004 µg kg$^{-1}$; and Hg, 0.0003 µg kg$^{-1}$. Trace metal concentrations were compared to the ER-L (Effect Range-Low) and ER-M (Effect Range-Median)

values reported in the sediment guidelines of the U.S. Environmental Protection Agency (USEPA) (Long et al., 1995). Subsamples (30–40 mg) were oven-dried at 50 °C and were pulverized to a silt size using an agate mortar; 10 mg of each of these samples were enclosed in thin Ag film cups. Next, 1 M HCl was added to these samples, and these were dried at 11 °C for 30 min. TOC, TN, and TS contents were then measured with an elemental analyzer (Flash 2000, Thermo Scientific, Italy). Each sediment core sample was oven-dried at 60 °C immediately after the fieldwork. After the shells had been removed, the

samples were homogenized using a mortar and pestle. After homogenization, 3–5 g subsamples were placed in pre-weighed crucibles that were then placed in a desiccator overnight to remove any remaining moisture. The samples were weighed before being placed in a small furnace for 4 h at 550 °C and were then cooled in a desiccator overnight and reweighed. Loss on ignition (LOI) was calculated as follows: LOI% = [(Initial Dry Mass − Final Dry Mass)/Initial Dry Mass] × 100 (modified from Dean, 1974).

The sediment pH was measured using a pH Spear (Oakton, Eutech Instruments, Singapore) with a pH range of −1.00 to 15.00 pH and a resolution of 0.01 pH.

### 3.3 $^{210}$Pb radioactivity

With a nearly uniform grain size, the sediment composition of the three cores may indicate that trace metals have accumulated in a very stable sedimentary environment. In this case, the pollution history of these sediments can be successfully

reconstructed using $^{210}$Pb radiometric dating alone without normalizing for the concentrations of pollutant metals or referring to other time markers (Kitano et al., 1980; Grousset et al., 1999). Once heavy metals are released into shallow marine environments, they are removed from the water column through interactions with suspended particles and are subsequently deposited as bottom sediments (Callender, 2003). Since heavy metals deposited in sediments are not biodegradable, the metal profiles of the sediment cores, in combination with $^{210}$Pb dating techniques, can be used as records of pollution events (Cantwell

et al., 2007; Ip et al., 2007; Irabien et al., 2008; Lim et al., 2012).



Measurements of $^{210}$Pb, which has a 22.3-year half-life, were obtained from the sediment core by the Korea Basic Science Institute. The total $^{210}$Pb activity was determined based on the quantification of deposition of the granddaughter isotope $^{210}$Po on an Ag disc in conjunction with a $^{209}$Po chemical tracer (Ruiz-Fernández et al., 2003; Lubis, 2006).

### 3.4 Foraminiferal analysis

An analysis of the benthic foraminifera in core sediments was conducted at 2-cm intervals. Samples for this analysis were washed over a 63-µm sieve and were oven-dried at 50 °C. After further drying, these samples were subdivided using a modified Otto microsplitter. Foraminifera were counted under a binocular microscope from a known fraction, or the full sample was counted. A minimum of 200 individuals were counted from each interval. The benthic foraminifera taxonomy used in this paper is based on works by Asano (1950, 1951a, b, 1952), Matoba (1970), and Loeblich and Tappan (1994).

The total numbers of benthic foraminifera, the numbers of individuals per 20 mL, species diversity, and species evenness were statistically analyzed. Species diversity (H') and evenness (J) were calculated using formulas presented by Shannon and Weaver (1963) and Pielou (1966).

To determine the structure of the foraminiferal data set, we performed Q-mode clustering techniques with the paired group algorithm based on Bray–Curtis similarity which provided grouped averaged data for square-root-transformed abundance data. All statistical analyses were performed on foraminiferal relative abundance data sets using all species. Q-mode cluster analyses were carried out using the PRIMER 6 software (Plymouth Routines in Multivariate Ecological Research, UK). A correlation matrix was calculated for transformed geochemical elements and dominant species.

For the structural refinement, principal component analysis (PCA) was conducted to identify similarities and differences among foraminiferal assemblages. This technique reduces large data matrices composed of several variables to a small number of factors that represent the main modes of variation, facilitating the interpretation of large volumes of data. PCA was carried out for the ordination of sample locations based on the matrix constructed using 14 variables (geochemical elements).

## 4 Results

### 4.1 Grain size, geochemical composition, and trace metal contents

Sediments of core 11257 were composed mainly of homogeneous mud facies with 68.84% clay and 30.02% silt (Table 1, Figure 2-A). The OM content averaged 7.25% with a range from 6.53 to 8.29% (Table 1). The TOC and TN content averaged 0.9% and 0.15% with ranges from 0.77 to 1.08% and 0.08 to 0.20%, respectively (Table 1), and both gradually increased from the lowermost to the uppermost layers of the cores (Figure 2-B, C). The TS content, pH, carbon-to-nitrogen ratio (C/N), and carbon-to-sulfur ratio (C/S) averaged 0.15%, 6.97, 6.05, and 6.33 with ranges of 0.08–0.22%, 6.75–7.24, 5.03–10.49, and 4.41–12.12, respectively (Table 1). The average concentrations of the trace metals in core 11257 were below the ER-L values, except for Ni with a concentration of 23.5 mg/kg (Table 1). The concentration of Mn, Zn, Pb, and Cu, 468.9 mg/kg, 52.9



mg/kg, 15.8 mg/k and 0.07 mg/kg on average, respectively, increased slightly from the lowermost to the uppermost layers (Figure 2-E~H).

Sediments of core 10863 were composed mainly of homogeneous mud facies with 66.10% clay and 33.25% silt (Table 2, Figure 2-I). The OM content, TS content, pH, and C/N averaged 7.62%, 0.26%, 6.94, and 5.44 with ranges of 6.74–8.69%, 0.19–0.3%, 6.74–7.13, and 4.82–6.13, respectively (Table 2). The TOC and TN content averaged 1.07% (range, 0.82–1.27%) and 0.2% (range, 0.15–0.26%), respectively, and increased gradually from the lower layers of the core upward (Figure 2-J, K). However, the TOC content increased rapidly from a depth of 21 cm and appeared to remain constantly to the uppermost layer (1.22% on average). The C/S averaged 4.16, and was mostly distributed between 3.0 and 5.0 (Table 2). The average concentrations of the trace metals in core 10863 were below the ER-L values, except for Ni with a concentration of 23.4 mg/kg (Table 2). The concentration of Mn, Zn, Pb, and CU averaged 517.1 mg/kg, 76.1 mg/kg, 17.5 mg/k, and 0.09 mg/kg, respectively, and increased slightly from the lowermost to the uppermost layers (Figure 2-M~P).

Sediments of core 11285 were mainly composed of homogeneous mud facies with 66.95% clay and 32.64% silt (Table 2, Figure 2-Q). The OM, TOC, TN and TS content averaged 8.76%, 1.07%, 0.28%, and 0.46% with ranges of 8.1–10.26%, 0.96–1.29%, 0.24–0.35%, and 0.32–0.62%, respectively, and the C/N, with an average value of 3.86, ranged from and 3.30 to 4.47 (Table 3). The C/S averaged 2.36 and ranged from 1.85 to 3.08; these values were generally below 3.0 throughout the core, except in the layers at depths of 5 cm, 57 cm, and 59 cm (Table 3). The C/S values declined from the lowest layer (3.07) to the layer at 39 cm of depth (2.02) (Figure 2-T), and maintained an average concentration of 2.10 until 7 cm in depth, although the TOC and TN content did not show distinct corresponding variations (Figure 2-R, S). For trace metal concentrations, Ni, with an average of 24.1 mg/kg, was the only metal to exceed its ER-L value, and the average concentrations of the remaining metals were all below the corresponding ER-L (Table 3) values. Variations in concentrations of the trace metals Mn, Zn, Pb, and Cu appeared at 37 cm in depth with a relatively broad distribution (Figure 2-U~X). The concentration of Mn, which averaged 560.57 mg/kg, decreased distinctly from the lowest layers (705.20 mg/kg) to a depth of 27 cm (434.80 mg/kg) and appeared constant from that level to the uppermost layer (485.77 mg/kg) (Figure 2-U). Concentrations of Zn, which averaged 99.39 mg/kg overall, varied at 37 cm of depth from an average of 93.18 mg/kg to an average of 102.98 mg/kg.

## 4.2 Benthic foraminifera

Sixty-six species of benthic foraminifera (four agglutinated, 56 calcareous-hyaline, and six calcareous-porcelaneous) belonging to 43 genera were identified from sediments of core 11257 (Appendix A). The abundance frequency of agglutinated foraminifera, 7.7% on average, gradually increased from the lowermost to the uppermost layers (Figure 3-I-A). The dominant species (over 10% of abundant frequency in one layer) of benthic foraminifera, out of an average total of 6,608 individual benthic foraminifera in 20 ml of sediment, were *Ammonia beccarii* (average: 14.0%), *Elphidium advenum* (average: 13.3%), *E. clavatum* (average: 13.1%), *E. subarcticum* (average: 12.4%) and *A. ketienziensis* (average: 6.5%). Therefore, the abundance frequencies among these dominant species differs very little. Variations in the abundance frequencies of these dominant species



showed no clear trends from the lowermost to the uppermost layers (Figure 3-I-B~F), and the species diversity was high with an average value of 2.8 and a range from 2.6 to 2.9, indicating no clear trends in variation. The total number of benthic foraminifera (TNBF) decreased gradually from the lowest layer (8,016 individuals) to 11 cm (4,288 individuals) and increased again to 5 cm (11,296 individuals) (Figure 3-I-H). A cluster analysis was conducted using the Bray-Curtis similarity index (SI)

to examine the similarities among the component species at each location in which benthic foraminifera were found in any given sample (Figure 4-A). Twenty samples were categorized into one cluster with similarity (SI) ≈ 74.51. Cluster I was composed of an *A. beccarii-E. advenum-E. clavatum* (Ab-Ea-Ec) assemblage.

Fifty-six species of benthic foraminifera (four agglutinated, 46 calcareous-hyaline, and 6 calcareous-porcelaneous) belonging to 39 genera were identified from sediments of core 10863 (Appendix B). The dominant species of benthic foraminifera, out

of a total average of 2,544 individual benthic foraminifera in 20 ml of sediment, were *E. subarcticum* (average: 20.9%), *A. beccarii* (average: 14.9%), *E. clavatum* (average: 12.5%), *E. advenum* (average: 11.8%) and *A. ketienziensis* (average 6.5%). Co-occurring species were *E. somaense* (average: 5.2%) and *B. frigida* (average: 3.8%). The abundance frequency of *E. subarcticum* sharply increased from 17 cm depth toward the uppermost layers (Figure 3-II-F), although the frequencies of *A. beccarii*, *A. ketienziensis*, *E. advenum* and *E. clavatum* showed no clear variation (Figure 3-II-B~E). Species diversity averaged

2.5 and gradually decreased from the 17 cm depth to the uppermost layer (Figure 3-II-G). The TNBF increased gradually from the lowermost layer (3,816 individuals) to the uppermost layer (1,920 individuals) (Figure 3-II-H). Based on the cluster analysis, twenty-two samples were categorized into two clusters with SI ≈ 72.0 (Figure 4-B). Cluster I, which consisted of samples from 0–17 cm core depth, was an *E. subarcticum-A. beccarii* (Es-Ab) assemblage with average abundance frequencies of 30.2–42.6% and 9.6–22.0%, respectively. Cluster II, which consisted of samples from 19–43 cm core depth, was an *A.*

*beccarii-E. advenum-E. clavatum* (Ab-Ea-Ec) assemblage with average abundance frequencies of 14.5%, 14.3% and 14.2%, respectively, notably with very little difference among these abundance frequencies.

Twenty-seven species (five agglutinated, 20 calcareous-hyaline, and two calcareous-porcelaneous) belonging to 22 genera were identified from sediments of core 11285 (Appendix C). The abundance frequency of agglutinated species averaged 13.3%, and these taxa consisted mainly of *Eggerella advena* and *Trochammina hadai*, and gradually increased in abundance from 39

25   cm depth to the uppermost layer (Figure 3-III-A). The dominant species were *E. subarcticum* (average: 31.8%), *A. beccarii* (average: 27.4%), *B. frigida* (average: 15.6%), *T. hadai* (average: 10.8%), *E. advenum* (average: 5.3%), *E. somaense* (average: 5.3%), *Eg. advena* (average: 4.0%) and *E. clavatum* (average: 3.5%). The abundance frequency of *E. subarcticum*, the most dominant species, rapidly increased from 27 cm depth to the uppermost layer (Figure 3-III-I). In contrast, the abundance frequencies of *A. beccarii* (Figure 3-III-D) and *E. advenum* (Figure 3-III-F) rapidly decreased upward from 41 and 45 cm

depth, respectively. The abundance frequency of *T. hadai* increased from 39 cm depth to the uppermost layer (Figure 3-III-C). The species diversity was very low with an average value of 1.5, and it showed no distinct variation pattern (Figure 3-III-J). The TNBF, with an average of 548 individuals, decreased gradually from the lowermost layer (1,160 individuals) to 11 cm depth (80 individuals), and reached its highest value (2,682 individuals) at 7 cm of depth (Figure 3-III-K).





In the results of the cluster analysis, 30 samples were categorized into three clusters with SI ≈ 62.56 (Figure 4-C). Cluster I, which consisted of only the sample from 35 cm depth was a *B. frigida-A. beccarii-E. subarcticum* (Bf-Ab-Es) assemblage with average abundance frequencies of 21.9%, 17.1% and 11.6%, respectively. Cluster II was composed of three sub-cluster (IIa, IIb, and IIc) with SI ≈ 72.77 (Figure 4-C). Cluster IIa consisted of samples from 11–15 cm and 29 cm depth (IIa1) as well as

samples from 39 cm and 41 cm depth (IIa2), with SI ≈ 74.0 (Figure 4-C). Cluster IIa1 was an *A. beccarii-E. subarcticum-T. hadai* (Ab-Es-Th) assemblage with abundance frequencies of 32.8%, 23.1% and 9.9%, respectively. Cluster IIa2 was an *E. somaense-A. beccarii-B. frigida* (Eso-Ab-Bf) assemblage with abundance frequencies of 23.7%, 20.7% and 19.9%, respectively. Cluster IIb, which was composed of samples from 31 and 33 cm depth, was a *T. hadai-E. subarcticum-B. frigida* (Th-Es-Bf) assemblage with abundance frequencies of 33.3%, 20.8% and 19%, respectively. Cluster IIc consisted of samples

from 1–9 cm and 15–27 cm depth and was an *E. subarcticum* (Es) assemblage with an average abundance frequency of 64.9% (range: 46.1–74.3%). Cluster III consisted of samples from 37 cm and 43–59 cm depth, and was an *A. beccarii-B. frigida-E. advenum* (Ab-Bf-Ea) assemblage average abundance frequencies of 49.2%, 20.8% and 9.5%, respectively.

### 4.3 $^{210}$Pb ages of core sediments

The total $^{210}$Pb in the sediments of cores 11257, 10863 and 11285 ranged from 17.2 to 64.7 mBq/g, 20.2 to 89.7 mBq/g and

15     mBq/g to 99.5 mBq/g, respectively (Table 4). The excess or unsupported $^{210}$Pb values were determined by subtracting the supported $^{210}$Pb (according to the asymptotic value) from the total $^{210}$Pb measured at each depth. Excess $^{210}$Pb did not decline exponentially downward in the cores, and the highest $^{210}$Pb values of cores 10863 and 11285 were found at depths of 3 cm and 7 cm, respectively. These findings suggest that the sediment accumulation rates vary over time. Alternatively, the erratic distribution of $^{210}$Pb concentrations may be attributed to biological mixing. The constant rate of supply (CRS) model,

commonly used to derive $^{210}$Pb dates, was used to calculate sediment ages and sedimentation rates (Appleby and Oldfield, 1992). Based on the results of the CRS model, sediments of core 11257, 10863 and 11285 were approximately dated to the years 1908 (thickness: 40 cm), 1904 (thickness: 44 cm) and 1906 (thickness: 60 cm), respectively (Table 4). The uppermost limit of the sedimentation period, which was about 2014, was established, and the lowest limit was set to about 1904 based on the half-life (22.3 years) of $^{210}$Pb (Sanchez-Cabeza and Ruiz-Fernandez, 2012) to ensure the accuracy of data acquired during

the sedimentation periods of the three cores. The sedimentation rate of the core 11257 averaged 0.44 cm/year with a range from 0.31 to 0.84 cm/year, and it increased very slowly upward (Figure 5-A). The sedimentation rate of core 10863 averaged 0.60 cm/year with a range from 0.34 to 1.71 cm/year, and it increased gradually after 1990 (Figure 5-B). The sedimentation rate of core 11285 averaged 1.19 cm/year with a range from 0.41 to 4.40 cm/year, and it increased rapidly after 1960 (Figure 5-C).

**5 Discussion**



Sediments composed of homogeneous fine-grained mud facies with 30.02–33.25% silt and 66.10–68.84% clay accumulated in cores 11257 (western area), 10863 (eastern area), and 11285 (northwestern area) over about 110 years. These sediments, however, show mutually distinct differences in the variation profiles of the concentrations of trace metals and geochemical elements, as well as in statistical data of benthic foraminiferal assemblages (Tables 1, 2 and 3, Figs. 2 and 3). In particular, the

difference was more definite in the comparison of benthic foraminiferal assemblage in the three stations (Figs. 4 and 6). The western and eastern areas are respectively characterized by only one benthic foraminiferal assemblage, the Ab-Ea-Ec assemblage (Figure 6-A), and two benthic foraminiferal assemblages, the Ab-Ea-Ec assemblage (lower layers, 44–18 cm) and the Es-Ab assemblage (upper layers, 18–0 cm) (Figure 6-B). However, the benthic foraminiferal assemblage of the northwestern area progressed from an Ab-Bf-Ea assemblage in the lowermost layers to an Eso-Ab-Bf assemblage, Ab-Bf-Ea

assemblage, Bf-Ab-Es assemblage, Th-Es-Bf assemblage, Ab-Es-Th assemblage, and finally to Es assemblage in the uppermost layers (Figure 6-C). It may be appropriate to use the records of the northwestern area, which exhibits variation in the foraminiferal assemblages and geochemical elements, as the standard for variation in the pollution history of Gamak Bay over the last 110 years.

## 5.1 The pre-urbanization period

The Ab-Bf-Ea assemblage, which was deposited between 1906 and 1964 in the northwestern area is characterized by a high abundance frequency of *A. beccarii* with 49.2% (Figure 3-III-Zone A, Figure 6-C), and correlated with the Ab-Ea-Ec assemblage of lower layers (deposited between 1904 and 1988) in the eastern area (Figure 6-B) and the Ab-Ea-Ec assemblage (deposited between 1908 and 2014) of the western area (Figure 6-A). A. beccarii, which is a principal dominant species in all three areas, is broadly distributed around the inner bay, tidal flats, brackish-water environments of the Pacific Ocean (Murray,

1991; Alve and Murray, 1999; Hayward et al., 2004; Murray 2006), and the brackish coastal areas of the East China Sea and Huanghai (Yellow) Sea (Wang et al., 1985), as well as in Gyunggi Bay and Ansan Bay in South Korea (Chang and Lee, 1984; Woo and Lee, 2006). *B. frigida*, *E. advenum* and *E. clavatum* are widely distributed around the southern and western coastal areas of Korea (Woo and Lee, 2006; Lee et al., 2016). Lee et al. (2016) reported that *E. advenum* was positively correlated to dissolved oxygen, pH, and C/S in sediments of Gamak Bay. Therefore, this dominance in this taxon does not appear to be

distinct to the pollution phenomenon described herein. However, statistical data from benthic foraminifera with a species diversity of 1.37 and a TNBF of 704 individuals in the northwestern area indicates a progression in the degree of pollution in the sediment (Alve, 1995) compared to the eastern area with a TNBF of 3.068 individuals, a species diversity of 2.67, and western area with a TNBF of 6,608 individuals with species diversity of 2.8. C/S ratios greater than 5, between 3 and 5, and less than 3 indicate freshwater conditions, oxic marine-to-brackish conditions, and reductive brackish marine conditions,

respectively (Berner and Raiswell, 1984). The northwestern area with a C/S of 2.63 may be a somewhat brackish environment compared to the eastern area with a C/S of 3.73 and western area with a C/S of 6.33.





The Korean coastal zones began to undergo an urbanization in earnest in the 1910s with an increase in fossil fuel (especially coal) consumption (Jeong et al., 2006; Lim et al., 2012, 2013). Therefore, pollution or a reductive brackish environment in the northwestern area, even though there were no differences in species composition of benthic foraminifera, may be caused by stagnant sewage supplied from a small village that formed before city construction in the hinterland of the northwestern area,

and the oval-shaped bottom topography of northwestern area, as shown from the difference of the sedimentation rate between northwestern (0.64 cm/y) and eastern (0.42 c m/y), western (0.44 cm/y) area.

## 5.2 Urbanization period

Benthic foraminiferal assemblage in the northwestern area varied rapidly from the Eso-Ab-Bf assemblage, through Ab-Bf-Ea, Bf-Ab-Es, Th-Es-Bf, to Ab-Es-Th assemblage in a brief space of time between 1965 and 1987. The abundance frequency of

*T. hadai*, *E. somaense* and *E. subarcticum* increased compared to that during the pre-urbanization period, but *A. beccarii* decreased, and *B. frigida* did not show a remarkable variation (Figure 3-III-Zone B). This period is characterized by an increase in the *E. subarcticum*, which is known as a bioindicator of organic pollution in Gamak Bay, with a DO content of 0.4 mg/L at oxygen minimum zones (OMZs: Helly and Levin, 2004; Paulmier and Ruiz-Pino, 2008) (Lee et al., 2016) and agglutinated species including *T. hadai*, which increase in abundance under eutrophic conditions. The representative foraminiferal

assemblages from hypoxic sediments associated with eutrophication have been found in Osaka Bay, Japan (Tsujimoto et al., 2006a, 2006b, 2008) and Gamak Bay, South Korea (Lee et al., 2012, 2016). During deposition of this interval, the sedimentation rate was higher than that during the pre-urbanization period, i.e., 1.0 cm/year versus 0.64 cm/year, and C/S decreased from 2.63 to 2.08. TNBF decreased relative to the pre-urbanization period from 704 to 244 individuals. Statistic indices of geochemical and benthic foraminiferal assemblages still indicate unfavorable sediment and environmental

conditions. A rapid variation in the benthic foraminiferal assemblages may reflect the unstable habitat environment that was caused by continuous inflow and deposition of sewage into sediment, as shown with an increase in the abundance frequency in *T. hadai* and *E. subarcticum*.

This sewage is thought to have caused eutrophication, and the water quality deteriorated more than during the pre-urbanization period. These conditions may have been caused by the increase in sewage discharged from Yeosu City, which was constructed

at the hinterland of northernmost area in Gamak Bay for people employed at the Yeocheon Industrial Complex from the late 1960s to the early 1970s (Kim et al., 2014).

The dense population and associated rapid urbanization that emerged in the 1960s resulted in great ecological stresses affecting the coastal ecosystems along the Korean coast (Choi et al., 2010). No benthic foraminiferal assemblages in the eastern and western areas of the bay correlate with the Eso-Ab-Bf ~ Ab-Es-Th assemblages of the northwestern area, and these findings

suggest that pollution during the urbanization period may have been restricted to the northwestern area and to not diffuse to the surrounding area (Figure 6-C).



## 5.3 The aquaculture period

The Es, Ab-Es-Th and Es assemblages were sequentially distributed in the northwestern area (Figure 6-C) from 1988 to 2014, and these are characterized by a high abundance frequency of *E. subarcticum* (Figure 3-III-Zone C). The Es assemblage in sediments deposited between 1988 and 2002 is composed mostly of *E. subarcticum* with an average abundance frequency of

55.2%, which is a bioindicator of organic pollution in Gamak Bay. The high sedimentation rate in this period, 1.54 cm/year, must be noted. The northwestern area of Gamak Bay has been a primary site for suspended mussel farming (*Mytilus galloprovincialis*) since the 1980s. Mussel farms can produce biodeposits such as feces and pseudofeces at rates that reach 3000 metric tons·ha$^{-1}$·year$^{-1}$ (Grenz, 1989), as well as shell parts that build up at a rate of 10 cm·year$^{-1}$, resulting in changes to the seabed approximately 20 m from a farm's boundaries (Dählback and Gunnarsson, 1981; Matisson and Lindén, 1983).

Sedimentation rates were greater within the farm than at reference sites, which supports the theory that mussel farming increases sedimentation rates (Callier et al., 2006). Therefore, the high sedimentation rate may be mostly the result of mussel farming. It is widely accepted that the primary benthic environmental impact of suspended mussel farming is the buildup of biodeposits directly below the culture area (Jaramillo et al., 1992; Hargrave, 2003), which may cause negative effects on the coastal systems, such as eutrophication and hypoxia, which occur at <0.2 mg·L$^{-1}$ or mL·L$^{-1}$ O$_2$ (Rabalais et al., 1991; Breitburg

et al., 2001; Diaz and Rosenberg, 2008; Doney, 2010; Gilbert et al., 2010; Kalantzi et al., 2013). The TNBF of 265 individuals/year, species diversity of 1.24 and C/S of 2.12 indicate worsening of benthic ecology and the sedimentary environment during this period associated with mussel farming.

The Ab-Es-Th assemblage distributed between 2003 and 2007 is composed of *A. beccarii* (abundance frequency: 35.2%), *E. subarcticum* (abundance frequency: 19.3%), *T. hadai* (abundance frequency: 17.1%), and has a TNBF of 99 individuals,

species diversity of 1.77, C/S of 2.10 and a sedimentation rate of 1.54 cm/year (Figure 6-C). During this period, statistical indices of benthic foraminifera and geochemical data indicate the deterioration of the benthic environment similar to the Es assemblage between 1988 and 2002. However, the difference in species composition between these two periods is associated with a rapid increase in the abundance frequency of *A. beccarii*, which is a common species in the inner bay. This difference may have been caused by an improvement in the benthic environment via dredging of the polluted sediment within the mussel

farm, which was conducted as part of the "Establishment of action plans for model coastal environmental management areas" (Ministry of Oceans and Fisheries, 2001).

The benthic foraminiferal assemblage varied to Es assemblage again after 2008 to 2014, which is composed mostly of *E. subarcticum*, with an average abundance frequency of 62.6%, TNBF of 1,323 individuals/year, species diversity of 1.23, C/S of 2.71 and a sedimentation rate of 3.15 cm/year (Figure 6-C). This interval has the highest abundance frequency of *E.*

*subarcticum* and TNBF after 1987. The abundance frequency of *A. beccarii* was the highest in the pre-pollution period, and it subsequently decreased rapidly. Conversely, the *E. subarcticum* abundance increased rapidly and reached its highest frequency during the aquaculture period (Figure 3-III-D, I). As the pollution increased, the populations of transitional or more tolerant species increased at the expense of taxa that are more sensitive, and highly tolerant or opportunistic species ultimately become





dominant (Alve, 1995). It is thought that organic pollution in the northwestern area of Gamak Bay progresses rapidly through a high sedimentation (rate: 3.15 cm/year) of biodeposits discharged from the suspended mussel farm. These polluted materials accumulated in sediment may contribute to the continuous deterioration and variation in the benthic ecological environment by means of "pollution storage," although these materials may be partially removed by residual bottom currents (Lee et al.,

2016).

The Es, Ab-Es-Th and Es assemblages of the northwestern area are correlated with the Es-Ab assemblage of the eastern area where oyster farming took place between 1989 and 2014. The latter assemblage is composed of *E. subarcticum* (35.4%) and *A. beccarii* (15.5%) and has a TNBF of 1787 individuals, species diversity of 2.18, C/S of 4.8 and a sedimentation rate of 0.95 cm/year (Figure 6-B). The environment conditions of the benthic ecology in the eastern area appear to have been better than

those in the northwestern area, although oyster farming also affects sedimentation and associated infaunal assemblages beneath cultivation areas (De Grave et al., 1998; Kaiser et al., 1998; Forrest and Creese, 2006; Dubois et al., 2007; Forrest et al., 2009; Solomon and Ahmed, 2016). In shallow-water eutrophic systems, the temporal and spatial changes in benthic foraminifera appear to be controlled primarily by the timing and extent of organic matter flux (Sabbatini et al., 2012). Organic matter in surface sediment is an important source of food for benthic fauna, but an overabundance may lead to a reduction in species

richness, abundance and biomass because of oxygen depletion and buildup of toxic byproducts, which may lead to an anaerobic and ultimately azoic state in heavily farmed or depositional areas; remediation of highly enriched sediments may take several years (Pereira et al., 2004). In addition, the extent of the environmental impact of the aquaculture depends on the amount of nutrients and organic matter that is released as well as on the hydrodynamic processes, including waves, current activity, and water residence time (Ackefors and Enell, 1994; Wu, 1995; Aure et al., 2007; Duarte et al., 2008; Stevens et al., 2008;

Strohmeier et al., 2008). Ultimately, polluted sediment in the northwestern area may be caused by overabundance and excessive deposition of organic matter through over-supply from mussel farming as well as the oval-shaped bottom physiography and very slow current speed (Lee et al., 2009), even though the species composition is somewhat similar between northwestern and eastern areas.

During the transition from the pre-urbanization to the urbanization period and aquaculture period, the concentration of Mn

(average: 560.57 mg/kg) in the northwestern area rapidly decreased from the lowermost layer (705.2 mg/kg) to the layer at 25 cm depth (431.0 mg/kg) and uppermost layer (470.2 mg/kg) (Figure 2-U), in contrast to gradually increasing values in the eastern and western area (Figure 2-E, M). This prominent upward decrease in Mn content is ascribed to its higher mobility than Fe and S with redox change during anoxic sediment diagenesis (Emerson et al., 1979; Kersten and Forstner, 1986; Calvert and Pederson, 1993). Lee et al. (2016) suggested that extremely polluted reducing conditions with a high OM content (>12.0%) and OMZs with dissolved oxygen (DO) <0.4 mg·L$^{-1}$ formed below the mussel farms in the northwestern area of Gamak Bay,

and that these conditions are associated with the appearance of *E. subarcticum* as a bioindicator. The processes of variation in benthic foraminiferal assemblages from the Ab-Bf-Ea assemblage, through Eso-Ab-Bf, Ab-Bf-Ea, Bf-Ab-Es, Th-Es-Bf, to





Ab-Es-Th assemblages and ultimately to the Es assemblage may represent the transition from oxic to anoxic environmental conditions.

## 5.4 Pollution variation

The organic matter discussed in the present study plays an important role in controlling the composition of benthic foraminiferal assemblages. The OM content and quality can be considered an important limiting factor for foraminiferal development (Armynot du Châtelet et al., 2008), as indicated by the TOC content and the quality of the organic matter represented by the C/N ratio (Foster et al., 2012). A PCA and scatter diagram analysis was conducted to identify the relationships between the foraminiferal assemblages and geochemical characteristics (Figure 7), and the variations between the sampling depth and pollution intensity (Figure 8), respectively. The TOC content of core 11257 does not show a distinct relationship and trend in PCA (Figure 7-A) and in the scatter diagram analysis (Figure 8-A). However, the TOC content of core 10863 has positive relationship with *E. subarcticum* and negative relationships with *E. advenum*, *E. clavatum* and species diversity (Figure 7-B). These results show a distinct pollution phenomenon from Group I sediments of 43–23 cm depth toward Group II sediments of 21–1 cm of depth and an Es-Ab assemblage (Figure 8-B). The TOC content of core 11285 has positive relationship with *E. subarcticum*, *E. advena* and *T. hadai*, and negative relationship with *A. beccarii*, *E. advenum*, *E. clavatum* and Mn (Figure 7-C). It varies in three phase from Group I to Group III in scatter diagram analysis (Figure 8-C). Group I consists of sediments from 59–47 cm depth, group II of sediments from 43–33 cm depth, and group III of sediments from of 29–1 cm (except of 23 cm) depth, which indicate the variation from the pre-urbanization period to the urbanization period, and the aquaculture period.

## 6 Conclusion

Historical records of the effects of anthropogenic pollution on the benthic foraminifera over the last 110 years in Gamak Bay are summarized in Figure 9 based on three periods. In the period between 1906 and 1964, Gamak Bay may have remained mostly unpolluted by anthropogenic activities, which allowed the Ab-Ea-Ec assemblage to flourish, except in the northwestern area with Ab-Bf-Ea assemblage that gradually deteriorated under the effects of a concave bottom physiography, flux limited sea water movement and sewage flux from small streams of villages (Figure 9-A). With city construction for people employed at the industrial complex between 1965 and 1987, the sewage inflow into the northwestern area increased rapidly, the pollution continuously accelerated, and the benthic foraminiferal assemblage rapidly progressed from Eso-Ab-Bf assemblage, through Ab-Bf-Ea, Bf-Ab-Es, Th-Es-Bf, to Ab-Es-Th assemblage in a brief space of time (Figure 9-B) via unstable habitat conditions. Pollution during the urbanization period may have been contained within the northwestern area, and it did not diffuse to the surrounding area. However, the northwestern area between 1988 and 2014 became progressively intensified by sewage flux and mussel farming in, and the organic pollution rapidly increased by a high sedimentation of the biodeposits (Figure 9-C).

The benthic foraminiferal assemblage transitioned into an Es assemblage composed of *E. subarcticum*, known to the bioindicator of organic pollution in Gamak Bay. These processes of variation in benthic foraminiferal assemblages at northwestern area may represent the transition from oxic to anoxic environmental conditions. During this period, pollution extended to the eastern area which affected oyster farming, causing a shift to an Es-Ab assemblage (Figure 9-C). The western

area, however, was unpolluted over the last 110 years due to the clockwise movement of the sea water inflow from the bay mouth of the south. The western area is considered to show reference conditions of Gamak Bay for both environmental parameters and foraminiferal assemblages.

## Acknowledgements

This research was supported by the Basic Science Research Program through the National Research Foundation of Korea

(NRF), funded by the Ministry of Science, ICT & Future Planning (NRF-2017R1A2B1006247). This work was also supported by grants from the National Institute of Fisheries Science (R2017048). This research was also supported by the Golden Seed Project, Ministry of Agriculture, Food and Rural Affairs (MAFRA), Ministry of Oceans and Fisheries (MOF), Rural Development Administration (RDA), and Korea Forest Service (KFS).

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







Figure 1: Sample locations of sediment cores 11257 (western area), 10863 (eastern area) and 11285 (northwestern area), and bathymetric map in semi-closed Gamak Bay, South Korea.



Figure 2: Variation of sediment composition (A, I, Q), total organic carbon (TOC) (B, J, R), total nitrogen (TN) (C, K, S), total organic carbon/ total sulfur (C/S) (D, L, T), Mn (E, M, U), Zn (F, N, V), Pb (G, O, W) and Cu (H, P, X) of sediment cores 11257 (A-H), 10863 (I-P) and 11285 (Q-X) in three sediment cores (11257, 10863, 11285).




Figure 3: Variation of test composition (I-A, II-A, III-A), dominant species (*Eggerella advena*: II-B, *Trochammina hadai*: III-C, *Ammonia beccarii*: I-B, II-B, III-D, *Ammonia ketienziensis*: I-C, II-C, *Buccella advena*: III-E, *Elphidium advenum*: I-D, II-D, II-F, *E. clavtum*: I-E, II-E, III-G, *E. subarcticum*: I-F, II-F, III-I, *E. somaense*: III-H), species diversity (I-G, II-G, III-J) and total number of benthic foraminifera (I-H, II-H, III-K) in benthic foraminiferal assemblages of three sediment cores (11257, 10863, 11285). Note, Compo.: composition, C-H




Foram.: Calcareous-Hyaline Foraminifera, C-P Foram.: Calcareous-Porcelain Foraminifera, A Foram.: Agglutinated Foraminifera, *A.:* *Ammonia, E.: Elphidium, Eg.: Eggerella,* TNBF: total number of benthic foraminifera, Indi.,: individual.

Figure 4: Cluster analysis using Bray–Curtis similarity of benthic foraminiferal assemblages in (A) sediment core 11257, (B) sediment core 10863 and (C) sediment core 11285 taken from Gamak Bay. Ab: *Ammonia beccarii,* Bf: *Buccella frigida,* Ea: *Elphidium advenum*, Ec: *E. clavatum,* Eso: *E. somaense,* Es: *E. subarcticum,* Th: *Trochammina hadai.*



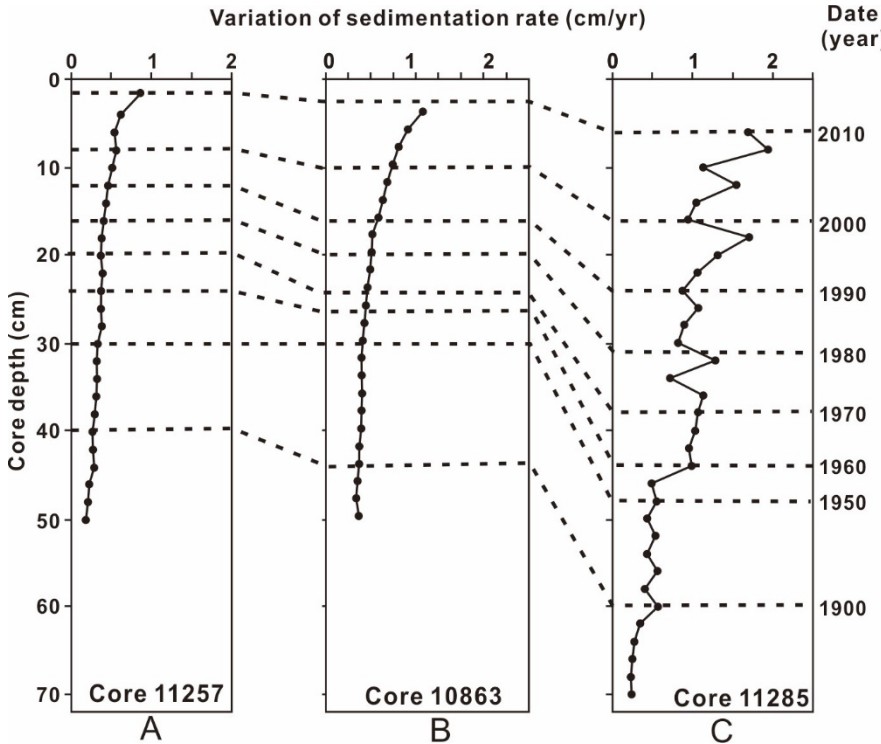

Figure 5: Variation of sediment age and sedimentation rate calculated from [210]Pb dates of sediment cores 11257 (A), 10863 (B) and 11285 (C), using the constant rate of supply (CRS) model (Appleby and Oldfield, 1992).





Figure 6: Correlation of benthic foraminiferal assemblages produced from (A) sediment core 11257, (B) sediment core 10863 and (C) sediment core 11285. Ab: *Ammonia beccarii,* Bf: *Buccella frigida,* Ea: *Elphidium advenum*, Ec: *E. clavatum,* Eso: *E. somaense,* Es: *E. subarcticum,* Th: *Trochammina hadai.* SR: sedimentation rate, TN: total nitrogen, C/S: total organic carbon/total sulfur, (H(S): species diversity, TNBF: total number of benthic foraminifera, Invi.: individual.



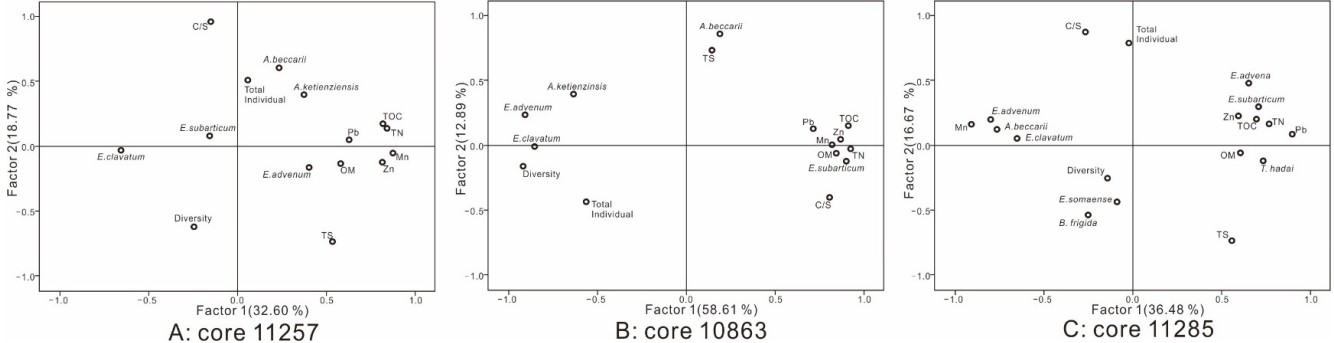

Figure 7: PCA (Principal Component Analysis) ordination diagram to statistic data of benthic foraminiferal assemblage and geochemistry in sediment cores 11257 (A), 10863 (B) and 11285 (C). OM: organic matter, TOC: total organic carbon, TN: total nitrogen, C/S: total organic carbon/total sulfur.

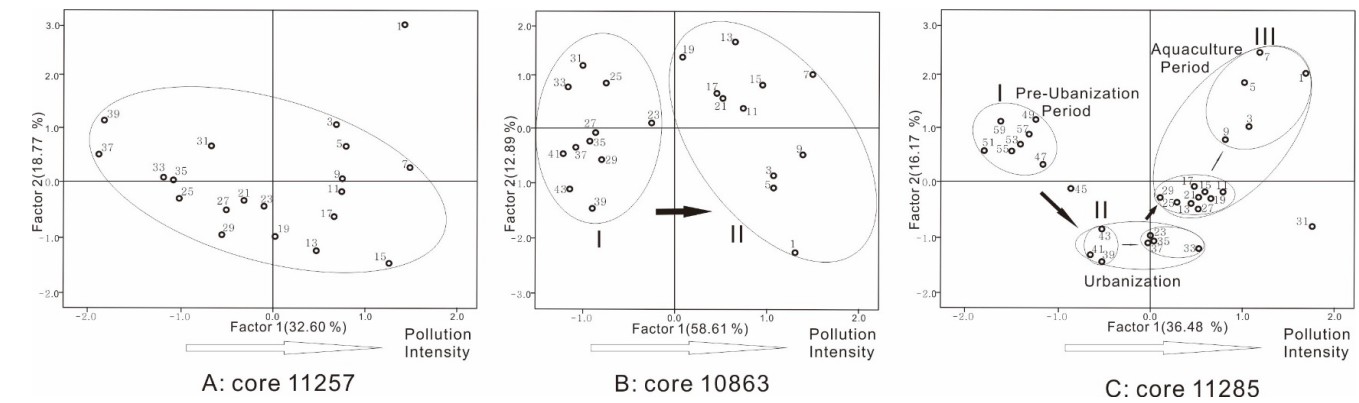

Figure 8: Scatter diagram analysis to depth of sediment cores 11257 (A), 10863 (B) and 11285 (C), based on statistic data of benthic foraminiferal assemblage and geochemistry.





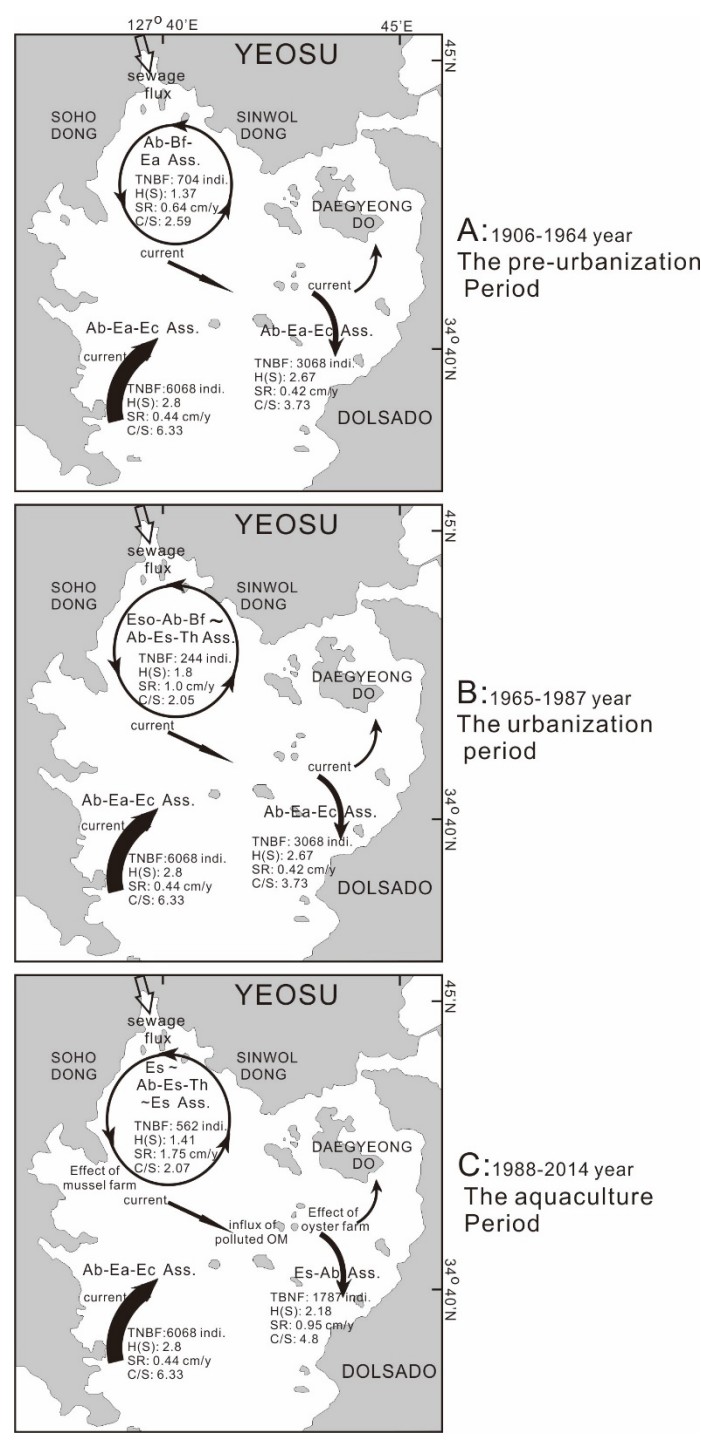

Figure 9: Synthesize mimetic diagram of the variation in historical records of the effects of anthropogenic pollution on benthic foraminifera over the last 110 years in Gamak Bay. Ab: *Ammonia beccarii,* Bf: *Buccella frigida,* Ea: *Elphidium advenum*, Ec: *E. clavatum,* Eso: *E. somaense,* Es: *E. subarcticum,* Th: *Trochammina hadai.* SR: sedimentation rate, OM: organic matter, C/S: total organic carbon/total sulfur, (H(S): species diversity, TNBF: total number of benthic foraminifera, Invi.: individual.





Table 1: Grain-size composition, trace metal and geochemical analysis of sediment core 11257, taken from western area of Gamak Bay. Note: Sedi: sediment, OM: organic matter, TOC: total organic carbon, TN: total nitrogen, TS: total sulfur, C/S: total organic carbon/total sulfur, C/N: total organic carbon/ total nitrogen.

| Depth (cm) | Trace Metal Content (mg/kg) | | | | | | | | | | | | Sedi. Composition (%) | | | | Sedi. Type | Mean (Ø) | pH | OM (%) | TOC (%) | TN (%) | TS (%) | C/N ratio | C/S ratio |
|---|---|---|---|---|---|---|---|---|---|---|---|---|---|---|---|---|---|---|---|---|---|---|---|---|---|
| | Al (%) | Fe (%) | Mn | Zn | Cr | Ni | Cu | Co | As | Cd | Pb | Hg | Gravel | Sand | Silt | Clay | | | | | | | | | |
| 1 | 3.17 | 1.91 | 512.20 | 55.60 | 48.20 | 23.94 | 13.19 | 11.10 | 4.34 | 0.05 | 16.86 | 0.01 | 0.13 | 2.49 | 33.38 | 64.00 | (g)sM | 9.48 | 6.75 | 7.35 | 1.08 | 0.20 | 0.09 | 5.49 | 12.12 |
| 3 | 3.11 | 1.88 | 457.80 | 60.60 | 47.40 | 23.17 | 31.15 | 10.80 | 4.50 | 0.06 | 16.28 | 0.02 | 0.07 | 2.34 | 35.16 | 62.43 | (g)sM | 9.33 | 6.75 | 7.77 | 0.98 | 0.19 | 0.11 | 5.25 | 8.86 |
| 5 | 3.12 | 1.93 | 478.80 | 52.80 | 46.60 | 24.29 | 11.87 | 10.86 | 5.04 | 0.06 | 16.21 | 0.02 | 0.00 | 2.49 | 32.70 | 64.81 | M | 9.51 | 6.80 | 8.29 | 0.97 | 0.17 | 0.17 | 5.53 | 5.57 |
| 7 | 3.54 | 2.00 | 513.60 | 57.60 | 52.60 | 24.21 | 13.06 | 10.89 | 4.81 | 0.07 | 26.65 | 0.01 | 0.00 | 1.62 | 36.66 | 61.72 | M | 9.36 | 6.83 | 6.58 | 0.90 | 0.18 | 0.15 | 5.03 | 5.84 |
| 9 | 3.28 | 1.98 | 499.20 | 55.00 | 48.60 | 23.60 | 11.44 | 11.21 | 4.61 | 0.06 | 16.15 | 0.02 | 0.00 | 1.48 | 14.20 | 84.32 | M | 10.17 | 6.86 | 7.00 | 1.01 | 0.17 | 0.18 | 5.84 | 5.59 |
| 11 | 3.01 | 1.93 | 482.80 | 54.20 | 45.80 | 23.89 | 11.28 | 11.27 | 4.52 | 0.06 | 15.88 | 0.02 | 0.00 | 1.13 | 35.01 | 63.86 | M | 9.48 | 6.87 | 7.15 | 0.93 | 0.17 | 0.19 | 5.34 | 4.93 |
| 13 | 3.05 | 1.92 | 468.00 | 56.80 | 45.80 | 23.61 | 20.16 | 10.75 | 4.29 | 0.06 | 16.23 | 0.02 | 0.00 | 1.06 | 3.97 | 94.97 | M | 10.77 | 6.87 | 7.36 | 0.96 | 0.17 | 0.21 | 5.76 | 4.54 |
| 15 | 3.40 | 2.01 | 507.00 | 56.60 | 50.40 | 24.75 | 12.71 | 11.37 | 4.12 | 0.13 | 16.41 | 0.02 | 0.00 | 0.76 | 30.58 | 68.66 | M | 9.82 | 6.88 | 8.14 | 0.99 | 0.18 | 0.22 | 5.52 | 4.41 |
| 17 | 3.44 | 2.02 | 505.80 | 55.60 | 51.20 | 23.96 | 12.13 | 11.00 | 4.25 | 0.06 | 15.80 | 0.01 | 0.00 | 0.92 | 34.55 | 64.53 | M | 9.42 | 6.91 | 8.07 | 0.92 | 0.17 | 0.20 | 5.48 | 4.71 |
| 19 | 3.36 | 2.00 | 483.00 | 53.60 | 50.00 | 23.66 | 12.08 | 10.74 | 4.32 | 0.06 | 15.29 | 0.01 | 0.00 | 0.93 | 19.99 | 79.08 | M | 9.79 | 6.91 | 7.23 | 0.87 | 0.15 | 0.17 | 5.75 | 5.09 |
| 21 | 3.27 | 1.94 | 466.00 | 51.20 | 48.20 | 23.41 | 10.02 | 11.07 | 4.29 | 0.06 | 14.84 | 0.01 | 0.00 | 0.91 | 23.88 | 75.21 | M | 9.64 | 6.92 | 7.18 | 0.84 | 0.14 | 0.16 | 5.81 | 5.38 |
| 23 | 3.39 | 1.99 | 466.60 | 53.40 | 50.20 | 24.16 | 10.36 | 11.21 | 4.40 | 0.06 | 15.10 | 0.02 | 0.00 | 0.85 | 31.40 | 67.75 | M | 9.52 | 6.96 | 7.07 | 0.97 | 0.16 | 0.16 | 6.16 | 6.07 |
| 25 | 3.40 | 1.97 | 462.80 | 51.80 | 49.80 | 23.66 | 11.30 | 10.89 | 4.42 | 0.07 | 14.97 | 0.02 | 0.00 | 0.80 | 34.57 | 64.63 | M | 9.57 | 7.01 | 6.75 | 0.83 | 0.08 | 0.14 | 10.49 | 5.84 |
| 27 | 3.24 | 1.90 | 453.00 | 54.60 | 47.80 | 23.42 | 20.05 | 10.92 | 4.35 | 0.06 | 14.70 | 0.01 | 0.00 | 0.64 | 35.08 | 64.28 | M | 9.50 | 7.06 | 7.31 | 0.82 | 0.14 | 0.13 | 6.03 | 6.37 |
| 29 | 3.49 | 1.99 | 468.60 | 53.60 | 51.00 | 24.08 | 12.75 | 11.12 | 4.29 | 0.06 | 15.14 | 0.02 | 0.00 | 0.65 | 33.69 | 65.66 | M | 9.66 | 7.10 | 7.10 | 0.83 | 0.15 | 0.17 | 5.58 | 4.79 |
| 31 | 3.45 | 1.91 | 451.40 | 50.40 | 51.20 | 23.30 | 9.92 | 10.46 | 4.26 | 0.08 | 14.68 | 0.01 | 0.00 | 0.65 | 33.25 | 66.10 | M | 9.66 | 7.12 | 7.41 | 0.84 | 0.14 | 0.12 | 5.82 | 7.08 |
| 33 | 3.26 | 1.88 | 438.80 | 47.40 | 47.80 | 22.59 | 10.86 | 10.69 | 4.29 | 0.06 | 14.64 | 0.01 | 0.00 | 0.79 | 35.36 | 63.85 | M | 9.46 | 7.12 | 6.82 | 0.80 | 0.13 | 0.13 | 6.14 | 6.39 |
| 35 | 2.99 | 1.83 | 426.20 | 46.80 | 45.40 | 22.47 | 9.68 | 10.38 | 4.12 | 0.06 | 13.97 | 0.01 | 0.00 | 0.72 | 35.86 | 63.42 | M | 9.46 | 7.15 | 7.27 | 0.83 | 0.14 | 0.14 | 5.90 | 6.06 |
| 37 | 3.13 | 1.83 | 412.40 | 46.00 | 47.00 | 22.23 | 9.06 | 10.28 | 4.13 | 0.06 | 13.66 | 0.01 | 0.00 | 0.69 | 26.02 | 73.29 | M | 9.50 | 7.21 | 6.53 | 0.78 | 0.10 | 0.11 | 8.10 | 7.38 |
| 39 | 2.90 | 1.79 | 424.60 | 45.00 | 44.40 | 21.60 | 9.43 | 10.23 | 3.96 | 0.05 | 13.94 | 0.01 | 0.00 | 0.66 | 35.07 | 64.26 | M | 9.47 | 7.24 | 6.53 | 0.77 | 0.13 | 0.08 | 5.88 | 9.52 |
| Min. | 2.90 | 1.79 | 412.40 | 45.00 | 44.40 | 21.60 | 9.06 | 10.23 | 3.96 | 0.05 | 13.66 | 0.01 | 0.00 | 0.64 | 3.97 | 61.72 | | 9.33 | 6.75 | 6.53 | 0.77 | 0.08 | 0.08 | 5.03 | 4.41 |
| Max. | 3.54 | 2.02 | 513.60 | 60.60 | 52.60 | 24.75 | 31.15 | 11.37 | 5.04 | 0.13 | 26.65 | 0.02 | 0.13 | 2.49 | 36.66 | 94.97 | | 10.77 | 7.24 | 8.29 | 1.08 | 0.20 | 0.22 | 10.49 | 12.12 |
| Avg. | 3.25 | 1.93 | 468.93 | 52.93 | 48.47 | 23.50 | 13.12 | 10.86 | 4.37 | 0.07 | 15.87 | 0.01 | 0.01 | 1.13 | 30.02 | 68.84 | | 9.63 | 6.97 | 7.25 | 0.90 | 0.15 | 0.15 | 6.05 | 6.33 |
| ER-L | N/A | N/A | N/A | 150.00 | 81.00 | 20.90 | 34.00 | N/A | 8.20 | 1.20 | 46.70 | 0.15 | | | | | | | | | | | | | |
| ER-M | N/A | N/A | N/A | 410.00 | 370.00 | 51.60 | 270.00 | N/A | 70.00 | 9.60 | 218.00 | 0.71 | | | | | | | | | | | | | |



Table 2: Grain-size composition, trace metal and geochemical analysis of sediment core 10863, taken from eastern area of Gamak Bay. Note: Sedi: sediment, OM: organic matter, TOC: total organic carbon, TN: total nitrogen, TS: total sulfur, C/S: total organic carbon/total sulfur, C/N: total organic carbon/ total nitrogen.

| Depth (cm) | Al (%) | Fe (%) | Mn | Zn | Cr | Ni | Cu | Co | As | Cd | Pb | Hg | Sand | Silt | Clay | Sedi. Type | Mean (Ø) | pH | OM (%) | TOC (%) | TN (%) | TS (%) | C/N ratio | C/S ratio |
|---|---|---|---|---|---|---|---|---|---|---|---|---|---|---|---|---|---|---|---|---|---|---|---|---|
| 1 | 3.59 | 2.37 | 565.00 | 81.20 | 47.60 | 23.46 | 16.18 | 10.38 | 4.11 | 0.08 | 18.93 | 0.02 | 0.54 | 31.02 | 68.44 | M | 9.68 | 6.74 | 8.30 | 1.24 | 0.26 | 0.19 | 4.82 | 6.54 |
| 3 | 3.58 | 2.35 | 616.20 | 86.40 | 49.40 | 24.76 | 24.43 | 10.54 | 4.48 | 0.09 | 18.97 | 0.02 | 0.89 | 30.55 | 68.56 | M | 9.64 | 6.75 | 7.68 | 1.25 | 0.25 | 0.22 | 4.93 | 5.68 |
| 5 | 3.57 | 2.39 | 621.40 | 100.00 | 48.80 | 24.58 | 42.93 | 10.51 | 4.65 | 0.09 | 18.81 | 0.02 | 0.50 | 29.33 | 70.17 | M | 9.75 | 6.75 | 7.92 | 1.22 | 0.23 | 0.26 | 5.27 | 4.74 |
| 7 | 3.76 | 2.48 | 641.20 | 101.00 | 50.20 | 26.54 | 43.23 | 10.88 | 4.67 | 0.09 | 19.53 | 0.02 | 0.38 | 30.31 | 69.31 | M | 9.80 | 6.76 | 8.48 | 1.23 | 0.23 | 0.28 | 5.27 | 4.40 |
| 9 | 3.51 | 2.32 | 569.80 | 93.40 | 47.40 | 23.97 | 41.07 | 10.34 | 4.23 | 0.09 | 18.39 | 0.02 | 1.32 | 36.68 | 62.00 | M | 9.38 | 6.80 | 8.69 | 1.24 | 0.22 | 0.25 | 5.52 | 4.89 |
| 11 | 3.41 | 2.21 | 518.20 | 104.20 | 44.20 | 22.66 | 66.34 | 9.67 | 4.26 | 0.24 | 17.56 | 0.02 | 0.49 | 32.90 | 66.61 | M | 9.60 | 6.81 | 8.17 | 1.26 | 0.23 | 0.29 | 5.46 | 4.36 |
| 13 | 3.55 | 2.30 | 545.00 | 76.20 | 45.60 | 23.40 | 15.57 | 10.25 | 4.47 | 0.08 | 18.96 | 0.02 | 1.33 | 32.26 | 66.41 | M | 9.54 | 6.83 | 8.28 | 1.20 | 0.22 | 0.27 | 5.44 | 4.39 |
| 15 | 3.73 | 2.41 | 575.00 | 81.60 | 50.40 | 24.66 | 16.55 | 10.74 | 4.73 | 0.09 | 19.38 | 0.02 | 0.64 | 32.39 | 66.98 | M | 9.59 | 6.83 | 8.00 | 1.22 | 0.22 | 0.28 | 5.56 | 4.31 |
| 17 | 3.47 | 2.37 | 572.00 | 78.80 | 47.40 | 23.73 | 15.57 | 10.50 | 4.40 | 0.09 | 18.55 | 0.02 | 0.43 | 30.63 | 68.94 | M | 9.78 | 6.86 | 7.77 | 1.15 | 0.21 | 0.29 | 5.45 | 3.93 |
| 19 | 3.72 | 2.22 | 526.00 | 70.20 | 46.40 | 22.29 | 14.46 | 9.73 | 4.23 | 0.09 | 17.14 | 0.02 | 0.61 | 33.09 | 66.29 | M | 9.56 | 6.88 | 7.84 | 1.19 | 0.21 | 0.27 | 5.78 | 4.41 |
| 21 | 3.72 | 2.22 | 528.80 | 100.80 | 52.20 | 24.66 | 66.19 | 9.62 | 4.00 | 0.09 | 25.53 | 0.02 | 0.27 | 33.21 | 66.52 | M | 9.53 | 6.92 | 7.89 | 1.27 | 0.21 | 0.30 | 6.13 | 4.28 |
| 23 | 3.65 | 2.32 | 535.80 | 74.40 | 48.60 | 24.40 | 16.18 | 10.31 | 4.18 | 0.09 | 17.60 | 0.02 | 0.40 | 34.44 | 65.17 | M | 9.47 | 6.95 | 7.41 | 1.09 | 0.19 | 0.29 | 5.73 | 3.82 |
| 25 | 3.37 | 2.22 | 497.60 | 69.80 | 45.40 | 22.56 | 12.67 | 9.94 | 4.14 | 0.08 | 16.49 | 0.02 | 0.70 | 35.10 | 64.20 | M | 9.48 | 6.98 | 7.05 | 1.01 | 0.19 | 0.29 | 5.39 | 3.43 |
| 27 | 3.37 | 2.24 | 481.40 | 66.00 | 44.80 | 22.74 | 11.22 | 9.92 | 4.37 | 0.07 | 16.12 | 0.02 | 0.23 | 35.83 | 63.94 | M | 9.41 | 7.02 | 6.94 | 0.94 | 0.17 | 0.26 | 5.36 | 3.65 |
| 29 | 3.56 | 2.22 | 469.00 | 64.00 | 46.00 | 22.61 | 10.81 | 10.10 | 4.78 | 0.07 | 16.36 | 0.02 | 0.71 | 34.66 | 64.63 | M | 9.58 | 7.05 | 7.54 | 0.91 | 0.18 | 0.25 | 5.18 | 3.70 |
| 31 | 3.20 | 2.11 | 439.00 | 59.80 | 42.40 | 21.76 | 12.60 | 9.55 | 4.63 | 0.07 | 15.13 | 0.02 | 1.16 | 36.48 | 62.36 | M | 9.35 | 7.07 | 6.86 | 0.94 | 0.17 | 0.24 | 5.59 | 3.99 |
| 33 | 3.20 | 2.18 | 453.80 | 61.00 | 42.40 | 22.67 | 11.07 | 9.93 | 4.74 | 0.07 | 15.63 | 0.02 | 0.38 | 35.05 | 64.57 | M | 9.50 | 7.08 | 6.74 | 0.86 | 0.16 | 0.25 | 5.49 | 3.48 |
| 35 | 3.59 | 2.26 | 484.60 | 61.60 | 46.40 | 23.78 | 10.12 | 10.39 | 4.79 | 0.07 | 15.67 | 0.02 | 0.21 | 37.07 | 62.72 | M | 9.40 | 7.08 | 6.97 | 0.85 | 0.16 | 0.27 | 5.21 | 3.12 |
| 37 | 3.13 | 2.15 | 462.60 | 59.00 | 41.20 | 21.97 | 9.38 | 9.61 | 4.12 | 0.06 | 15.11 | 0.01 | 1.10 | 35.82 | 63.08 | M | 9.44 | 7.10 | 7.00 | 0.82 | 0.15 | 0.24 | 5.32 | 3.43 |
| 39 | 3.45 | 2.22 | 495.80 | 63.20 | 46.60 | 23.48 | 11.08 | 10.17 | 4.21 | 0.07 | 15.69 | 0.02 | 0.61 | 26.60 | 72.79 | M | 9.61 | 7.10 | 8.09 | 0.87 | 0.15 | 0.23 | 5.71 | 3.79 |
| 41 | 3.27 | 2.22 | 298.60 | 63.40 | 44.40 | 23.55 | 11.07 | 10.16 | 4.27 | 0.07 | 15.78 | 0.02 | 0.58 | 33.21 | 66.22 | M | 9.39 | 7.13 | 7.22 | 0.88 | 0.16 | 0.25 | 5.65 | 3.49 |
| 43 | 2.87 | 2.13 | 479.40 | 59.00 | 39.80 | 21.81 | 9.77 | 9.68 | 3.92 | 0.06 | 15.16 | 0.02 | 0.70 | 34.96 | 64.33 | M | 9.47 | 7.13 | 6.79 | 0.86 | 0.16 | 0.23 | 5.44 | 3.70 |
| Min. | 2.87 | 2.11 | 298.60 | 59.00 | 39.80 | 21.76 | 9.38 | 9.55 | 3.92 | 0.06 | 15.11 | 0.01 | 0.21 | 26.60 | 62.00 | | 9.35 | 6.74 | 6.74 | 0.82 | 0.15 | 0.19 | 4.82 | 3.12 |
| Max. | 3.76 | 2.48 | 641.20 | 104.20 | 52.20 | 26.54 | 66.34 | 10.88 | 4.79 | 0.24 | 25.53 | 0.02 | 1.33 | 37.07 | 72.79 | | 9.80 | 7.13 | 8.69 | 1.27 | 0.26 | 0.30 | 6.13 | 6.54 |
| Avg. | 3.47 | 2.27 | 517.10 | 76.14 | 46.25 | 23.46 | 22.21 | 10.13 | 4.38 | 0.09 | 17.57 | 0.02 | 0.64 | 33.25 | 66.10 | | 9.54 | 6.94 | 7.62 | 1.07 | 0.20 | 0.26 | 5.44 | 4.16 |
| ER-L | N/A | N/A | N/A | 150.00 | 81.00 | 20.90 | 34.00 | N/A | 8.20 | 1.20 | 46.70 | 0.15 | | | | | | | | | | | | |
| ER-M | N/A | N/A | N/A | 410.00 | 370.00 | 51.60 | 270.00 | N/A | 70.00 | 9.60 | 218.00 | 0.71 | | | | | | | | | | | | |




Table 3: Grain-size composition, trace metal and geochemical analysis of sediment core 11285, taken from northwestern area of Gamak Bay. Note: Sedi: sediment, OM: organic matter, TOC: total organic carbon, TN: total nitrogen, TS: total sulfur, C/S: total organic carbon/total sulfur, C/N: total organic carbon/ total nitrogen.

| Depth (cm) | Al (%) | Fe (%) | Mn | Zn | Cr | Ni | Cu | Co | As | Cd | Pb | Hg | Sand | Silt | Clay | Sedi. Type | Mean (Ø) | pH | OM (%) | TOC (%) | TN (%) | TS (%) | C/N ratio | C/S ratio |
|---|---|---|---|---|---|---|---|---|---|---|---|---|---|---|---|---|---|---|---|---|---|---|---|---|
| 1 | 3.85 | 2.74 | 470.20 | 119.40 | 51.20 | 31.11 | 57.61 | 13.13 | 4.83 | 0.69 | 22.06 | 0.03 | 0.67 | 34.38 | 64.96 | M | 9.36 | 6.66 | 10.26 | 1.18 | 0.35 | 0.40 | 3.38 | 2.99 |
| 3 | 4.23 | 2.83 | 491.20 | 100.00 | 53.60 | 31.62 | 22.58 | 13.71 | 5.08 | 0.20 | 22.93 | 0.02 | 0.40 | 36.71 | 62.90 | M | 9.17 | 6.73 | 9.70 | 1.15 | 0.32 | 0.43 | 3.58 | 2.70 |
| 5 | 3.88 | 2.81 | 501.00 | 110.40 | 52.40 | 31.28 | 39.09 | 13.24 | 5.28 | 0.19 | 22.71 | 0.02 | 0.21 | 35.84 | 63.95 | M | 9.26 | 6.81 | 9.16 | 1.19 | 0.33 | 0.39 | 3.59 | 3.08 |
| 7 | 3.98 | 2.88 | 537.40 | 102.00 | 52.20 | 31.54 | 20.48 | 13.50 | 5.43 | 0.18 | 24.58 | 0.02 | 0.44 | 38.01 | 61.55 | M | 9.12 | 6.86 | 8.12 | 1.16 | 0.30 | 0.45 | 3.81 | 2.56 |
| 9 | 4.02 | 2.87 | 485.20 | 102.80 | 54.20 | 31.73 | 22.54 | 13.27 | 5.63 | 0.19 | 23.13 | 0.02 | 0.35 | 35.05 | 64.60 | M | 9.35 | 6.86 | 8.23 | 1.06 | 0.30 | 0.48 | 3.54 | 2.22 |
| 11 | 4.11 | 2.93 | 469.20 | 104.80 | 55.60 | 31.62 | 20.29 | 13.47 | 5.49 | 0.21 | 23.24 | 0.02 | 0.27 | 33.38 | 66.35 | M | 9.42 | 6.87 | 8.76 | 1.05 | 0.30 | 0.49 | 3.46 | 2.16 |
| 13 | 3.85 | 2.80 | 450.40 | 100.60 | 51.80 | 30.26 | 22.38 | 13.30 | 4.83 | 0.19 | 22.40 | 0.02 | 0.21 | 34.06 | 65.73 | M | 9.34 | 6.89 | 8.68 | 1.06 | 0.30 | 0.52 | 3.54 | 2.05 |
| 15 | 3.98 | 2.89 | 482.20 | 105.20 | 55.20 | 32.56 | 22.13 | 13.83 | 4.81 | 0.24 | 23.11 | 0.03 | 0.09 | 33.94 | 65.97 | M | 9.33 | 6.92 | 9.00 | 1.07 | 0.30 | 0.51 | 3.60 | 2.09 |
| 17 | 4.14 | 2.87 | 523.20 | 107.20 | 55.60 | 31.27 | 28.01 | 13.32 | 4.76 | 0.21 | 22.78 | 0.02 | 0.10 | 32.63 | 67.27 | M | 9.43 | 6.92 | 8.68 | 1.09 | 0.27 | 0.51 | 3.97 | 2.13 |
| 19 | 4.30 | 2.88 | 484.40 | 101.40 | 57.00 | 30.97 | 19.74 | 13.12 | 4.86 | 0.18 | 22.42 | 0.03 | 0.15 | 34.19 | 65.66 | M | 9.45 | 6.94 | 9.36 | 1.07 | 0.26 | 0.52 | 4.14 | 2.04 |
| 21 | 3.83 | 2.73 | 493.20 | 102.60 | 52.20 | 29.55 | 33.43 | 12.42 | 4.47 | 0.16 | 20.92 | 0.02 | 0.11 | 32.53 | 67.37 | M | 9.44 | 6.94 | 9.13 | 1.05 | 0.27 | 0.48 | 3.93 | 2.21 |
| 23 | 3.87 | 2.73 | 496.40 | 92.20 | 52.20 | 29.25 | 19.21 | 12.57 | 4.40 | 0.15 | 21.23 | 0.02 | 0.23 | 31.92 | 67.85 | M | 9.47 | 6.96 | 8.85 | 0.98 | 0.26 | 0.51 | 3.80 | 1.95 |
| 25 | 3.80 | 2.77 | 431.00 | 95.20 | 52.60 | 29.90 | 17.82 | 12.78 | 4.35 | 0.17 | 20.79 | 0.02 | 0.10 | 33.23 | 66.67 | M | 9.42 | 6.96 | 8.36 | 1.03 | 0.27 | 0.47 | 3.89 | 2.18 |
| 27 | 4.31 | 2.87 | 434.80 | 100.80 | 59.80 | 31.51 | 16.55 | 13.11 | 4.24 | 0.26 | 21.71 | 0.03 | 0.51 | 33.73 | 65.76 | M | 9.32 | 6.96 | 8.60 | 1.09 | 0.26 | 0.50 | 4.14 | 2.19 |
| 29 | 4.48 | 2.92 | 491.20 | 104.00 | 59.40 | 31.54 | 19.59 | 13.19 | 4.25 | 0.23 | 22.39 | 0.02 | 0.21 | 34.53 | 65.26 | M | 9.29 | 6.96 | 8.30 | 1.14 | 0.30 | 0.55 | 3.76 | 2.06 |
| 31 | 4.12 | 2.81 | 456.80 | 97.80 | 55.40 | 30.53 | 16.75 | 12.52 | 4.04 | 0.20 | 20.93 | 0.03 | 0.09 | 30.87 | 69.05 | M | 9.57 | 6.97 | 8.90 | 1.29 | 0.35 | 0.62 | 3.68 | 2.08 |
| 33 | 4.67 | 2.93 | 563.60 | 103.60 | 61.60 | 30.91 | 19.41 | 12.99 | 4.62 | 0.24 | 22.05 | 0.03 | 0.08 | 32.71 | 67.21 | M | 9.50 | 6.97 | 9.58 | 1.11 | 0.26 | 0.50 | 4.34 | 2.20 |
| 35 | 4.12 | 2.80 | 509.20 | 102.60 | 54.40 | 29.47 | 25.65 | 12.48 | 4.58 | 0.19 | 20.15 | 0.03 | 0.36 | 32.51 | 67.12 | M | 9.50 | 6.99 | 8.96 | 1.02 | 0.31 | 0.55 | 3.30 | 1.85 |
| 37 | 3.74 | 2.68 | 580.40 | 104.00 | 51.80 | 27.75 | 34.29 | 12.01 | 4.06 | 0.14 | 19.43 | 0.02 | 0.46 | 34.26 | 65.29 | M | 9.36 | 6.99 | 9.10 | 1.05 | 0.30 | 0.50 | 3.51 | 2.11 |
| 39 | 3.96 | 2.68 | 660.80 | 86.00 | 52.80 | 27.19 | 12.91 | 11.50 | 4.05 | 0.10 | 18.84 | 0.03 | 0.35 | 34.45 | 65.19 | M | 9.21 | 6.99 | 8.75 | 1.00 | 0.26 | 0.50 | 3.90 | 2.02 |
| 41 | 3.94 | 2.72 | 601.60 | 92.80 | 53.20 | 27.80 | 19.16 | 11.93 | 3.89 | 0.11 | 19.59 | 0.02 | 0.13 | 35.87 | 64.00 | M | 9.20 | 7.00 | 8.45 | 1.06 | 0.25 | 0.48 | 4.17 | 2.20 |
| 43 | 4.16 | 2.76 | 640.00 | 91.60 | 55.60 | 28.09 | 13.20 | 12.07 | 3.99 | 0.12 | 19.70 | 0.03 | 0.03 | 32.99 | 66.99 | M | 9.37 | 7.01 | 9.12 | 1.09 | 0.26 | 0.49 | 4.25 | 2.22 |
| 45 | 4.13 | 2.76 | 682.20 | 97.20 | 56.00 | 27.71 | 23.37 | 11.71 | 4.20 | 0.42 | 19.42 | 0.02 | 0.01 | 16.50 | 83.49 | M | 9.34 | 7.03 | 8.63 | 1.05 | 0.25 | 0.47 | 4.16 | 2.26 |
| 47 | 4.54 | 2.71 | 639.80 | 90.40 | 58.60 | 27.01 | 20.99 | 11.50 | 4.13 | 0.10 | 19.00 | 0.02 | 0.03 | 28.28 | 71.69 | M | 8.97 | 7.05 | 8.10 | 1.01 | 0.26 | 0.45 | 3.85 | 2.27 |
| 49 | 4.33 | 2.77 | 694.40 | 88.00 | 56.20 | 27.97 | 12.55 | 12.08 | 4.09 | 0.09 | 19.17 | 0.02 | 0.07 | 30.17 | 69.76 | M | 9.35 | 7.05 | 8.35 | 1.14 | 0.26 | 0.39 | 4.47 | 2.90 |
| 51 | 4.06 | 2.74 | 721.40 | 86.40 | 54.60 | 28.01 | 11.95 | 11.98 | 4.10 | 0.09 | 18.65 | 0.02 | 2.09 | 30.94 | 66.97 | M | 9.20 | 7.06 | 8.45 | 0.96 | 0.24 | 0.39 | 4.01 | 2.48 |
| 53 | 4.50 | 2.80 | 723.80 | 93.00 | 61.40 | 29.48 | 20.52 | 12.23 | 4.22 | 0.13 | 18.87 | 0.02 | 1.45 | 31.46 | 67.09 | M | 9.21 | 7.08 | 8.41 | 1.02 | 0.26 | 0.38 | 3.87 | 2.68 |
| 55 | 4.01 | 2.78 | 710.20 | 95.20 | 55.60 | 28.05 | 26.98 | 11.99 | 4.15 | 0.12 | 18.40 | 0.02 | 1.90 | 30.62 | 67.47 | M | 9.15 | 7.09 | 8.22 | 1.00 | 0.25 | 0.37 | 3.99 | 2.72 |
| 57 | 4.21 | 2.69 | 686.60 | 116.80 | 56.80 | 27.17 | 64.97 | 11.85 | 4.04 | 0.10 | 17.68 | 0.02 | 0.58 | 32.34 | 67.08 | M | 9.27 | 7.09 | 8.23 | 0.99 | 0.25 | 0.32 | 3.92 | 3.07 |
| 59 | 3.98 | 2.80 | 705.20 | 87.60 | 59.60 | 28.41 | 18.83 | 11.84 | 4.24 | 0.11 | 18.15 | 0.03 | 0.73 | 30.96 | 68.30 | M | 9.32 | 7.10 | 8.22 | 0.99 | 0.24 | 0.32 | 4.12 | 3.07 |
| Min. | 3.74 | 2.68 | 431.00 | 86.00 | 51.20 | 27.01 | 11.95 | 11.50 | 3.89 | 0.09 | 17.68 | 0.02 | 0.01 | 16.50 | 61.55 | | 8.97 | 6.66 | 8.10 | 0.96 | 0.24 | 0.32 | 3.30 | 1.85 |
| Max. | 4.67 | 2.93 | 723.80 | 119.40 | 61.60 | 32.56 | 64.97 | 13.83 | 5.63 | 0.69 | 24.58 | 0.03 | 2.09 | 38.01 | 83.49 | | 9.57 | 7.10 | 10.26 | 1.29 | 0.35 | 0.62 | 4.47 | 3.08 |
| Avg. | 4.10 | 2.80 | 560.57 | 99.39 | 55.29 | 29.71 | 24.10 | 12.62 | 4.50 | 0.19 | 20.88 | 0.03 | 0.41 | 32.64 | 66.95 | | 9.32 | 6.96 | 8.76 | 1.07 | 0.28 | 0.46 | 3.86 | 2.36 |
| ER-L | N/A | N/A | N/A | 150.00 | 81.00 | 20.90 | 34.00 | N/A | 8.20 | 1.20 | 46.70 | 0.15 | | | | | | | | | | | | |
| ER-M | N/A | N/A | N/A | 410.00 | 370.00 | 51.60 | 270.00 | N/A | 70.00 | 9.60 | 218.00 | 0.71 | | | | | | | | | | | | |





Table 4: Analysis results from the Constant Rate of Supply (CRS) model to calculate the age and sedimentation rates in sediment cores 11257 (A), 10863 (B) and 11285 (C), taken from Gamak Bay.

| Depth (cm) | Total $^{210}$Pb (mBq/g) | | | Supported $^{210}$Pb (mBq/g) | | | Excess $^{210}$Pb (mBq/g) | | | Dry mass (g) | | | Inventory Excess $^{210}$Pb(mBq/g) | | | Estimated Year | | | Date (year) | | | Accumulation rate (g/cm2/year) | | | Sedimentation rate (cm/year) | | |
|---|---|---|---|---|---|---|---|---|---|---|---|---|---|---|---|---|---|---|---|---|---|---|---|---|---|---|---|
| | A | B | C | A | B | C | A | B | C | A | B | C | A | B | C | A | B | C | A | B | C | A | B | C | A | B | C |
| 0-2 | 64.7 | 81.1 | 93.3 | 12 | 12 | 26 | 52.7 | 69.1 | 67.3 | 1.57 | 1.34 | 1.28 | 99.7 | 113.3 | 72.2 | 0.0 | 0.0 | 0.0 | 2014 | 2014 | 2014 | | | | | | |
| 2-4 | 59.9 | 89.7 | 86.9 | | | | 47.9 | 77.7 | 60.9 | 1.07 | 1.30 | 1.27 | 87.5 | 131.3 | 58.8 | 4.2 | 2.1 | 1.3 | 2010 | 2012 | 2013 | 0.76 | 1.44 | 2.12 | 0.84 | 1.71 | 4.40 |
| 4-6 | 46.6 | 85.1 | 95.3 | | | | 34.6 | 73.1 | 69.3 | 0.97 | 1.36 | 1.37 | 63.4 | 125.0 | 71.2 | 8.4 | 4.8 | 2.4 | 2006 | 2009 | 2012 | 0.57 | 0.97 | 1.73 | 0.62 | 1.13 | 3.37 |
| 6-8 | 36.3 | 84.0 | 99.5 | | | | 24.3 | 72.0 | 73.5 | 1.44 | 1.26 | 2.10 | 48.9 | 125.3 | 133.1 | 11.9 | 7.6 | 3.7 | 2002 | 2006 | 2010 | 0.55 | 0.82 | 1.52 | 0.54 | 0.94 | 1.68 |
| 8-10 | 47.5 | 78.7 | 97.4 | | | | 35.5 | 66.7 | 71.4 | 0.98 | 1.33 | 1.89 | 68.8 | 118.3 | 81.4 | 14.8 | 10.7 | 6.5 | 1999 | 2003 | 2008 | 0.55 | 0.73 | 1.10 | 0.56 | 0.83 | 1.93 |
| 10-12 | 46.5 | 82.0 | 88.9 | | | | 34.5 | 70.0 | 62.9 | 0.99 | 1.48 | 2.92 | 67.0 | 126.3 | 116.7 | 19.4 | 13.8 | 8.3 | 1995 | 2000 | 2006 | 0.50 | 0.68 | 1.06 | 0.52 | 0.75 | 1.14 |
| 12-14 | 43.8 | 80.9 | 82.8 | | | | 31.8 | 68.9 | 56.8 | 0.81 | 1.62 | 1.64 | 63.2 | 126.0 | 67.4 | 24.6 | 17.6 | 11.0 | 1989 | 1996 | 2003 | 0.46 | 0.63 | 0.92 | 0.46 | 0.68 | 1.55 |
| 14-16 | 40.6 | 75.8 | 79.7 | | | | 28.6 | 63.8 | 53.7 | 0.79 | 1.55 | 2.22 | 54.3 | 118.0 | 94.4 | 30.5 | 21.9 | 12.7 | 1983 | 1992 | 2001 | 0.43 | 0.58 | 0.92 | 0.45 | 0.62 | 1.05 |
| 16-18 | 37.1 | 72.6 | 78 | | | | 25.1 | 60.6 | 52 | 1.15 | 2.04 | 2.49 | 48.7 | 112.2 | 96.6 | 36.6 | 26.4 | 15.2 | 1977 | 1988 | 1999 | 0.40 | 0.54 | 0.87 | 0.41 | 0.58 | 0.94 |
| 18-20 | 22.7 | 64.3 | 74.1 | | | | 10.7 | 52.3 | 48.1 | 0.89 | 1.73 | 1.26 | 20.6 | 93.0 | 45.9 | 43.2 | 31.5 | 18.1 | 1971 | 1983 | 1996 | 0.38 | 0.50 | 0.81 | 0.39 | 0.56 | 1.70 |
| 20-22 | 22.8 | 61.5 | 72.8 | | | | 10.8 | 49.5 | 46.8 | 1.13 | 1.54 | 1.77 | 22.8 | 95.9 | 58.8 | 46.5 | 36.4 | 19.5 | 1968 | 1978 | 1994 | 0.39 | 0.48 | 0.83 | 0.37 | 0.49 | 1.31 |
| 22-24 | 25 | 59.4 | 71.9 | | | | 13.0 | 47.4 | 45.9 | 1.10 | 1.99 | 1.81 | 24.8 | 90.7 | 70.8 | 50.6 | 42.3 | 21.4 | 1963 | 1972 | 1993 | 0.39 | 0.45 | 0.82 | 0.41 | 0.47 | 1.06 |
| 24-26 | 20.6 | 56.4 | 73.9 | | | | 8.6 | 44.4 | 47.9 | 0.97 | 2.05 | 1.99 | 17.0 | 86.2 | 88.5 | 55.6 | 49.2 | 23.9 | 1958 | 1965 | 1990 | 0.38 | 0.42 | 0.80 | 0.38 | 0.43 | 0.87 |
| 26-28 | 18.5 | 45.1 | 75.1 | | | | 6.5 | 33.1 | 49.1 | 1.09 | 2.38 | 1.43 | 13.3 | 62.7 | 69.0 | 59.7 | 57.4 | 27.4 | 1954 | 1957 | 1987 | 0.38 | 0.39 | 0.76 | 0.38 | 0.41 | 1.08 |
| 28-30 | 25.9 | 38.5 | 67.8 | | | | 13.9 | 26.5 | 41.8 | 1.04 | 1.98 | 1.53 | 27.9 | 48.6 | 67.9 | 63.2 | 65.0 | 30.3 | 1951 | 1949 | 1984 | 0.39 | 0.37 | 0.73 | 0.39 | 0.40 | 0.90 |
| 30-32 | 24.2 | 32.4 | 66 | | | | 12.2 | 20.4 | 40 | 1.03 | 2.10 | 1.61 | 25.5 | 37.7 | 68.5 | 72.1 | 72.5 | 33.5 | 1942 | 1941 | 1981 | 0.36 | 0.35 | 0.71 | 0.35 | 0.38 | 0.83 |
| 32-34 | 17.7 | 22.1 | 63.5 | | | | 5.7 | 10.1 | 37.5 | 1.22 | 2.71 | 1.31 | 11.6 | 18.7 | 39.3 | 83.3 | 79.9 | 37.1 | 1931 | 1934 | 1977 | 0.33 | 0.34 | 0.68 | 0.33 | 0.37 | 1.30 |
| 34-36 | 17.5 | 21.3 | 63.1 | | | | 5.5 | 9.3 | 37.1 | 1.01 | 2.38 | 2.37 | 10.7 | 17.1 | 70.7 | 90.0 | 84.2 | 39.3 | 1924 | 1930 | 1975 | 0.33 | 0.34 | 0.68 | 0.34 | 0.37 | 0.72 |
| 36-38 | 16.4 | 20.6 | 61.7 | | | | 4.4 | 8.6 | 35.7 | 0.92 | 2.35 | 1.18 | 8.9 | 15.5 | 40.3 | 97.8 | 88.8 | 43.8 | 1916 | 1925 | 1970 | 0.32 | 0.34 | 0.64 | 0.31 | 0.38 | 1.14 |
| 38-40 | 17.2 | 23.6 | 63.7 | | | | 5.2 | 11.6 | 37.7 | 1.17 | 2.37 | 1.40 | 10.3 | 21.7 | 44.9 | 106.1 | 93.6 | 46.6 | 1908 | 1920 | 1967 | 0.31 | 0.34 | 0.63 | 0.31 | 0.36 | 1.07 |
| 40-42 | | 21.8 | 50.3 | | | | | 9.8 | 24.3 | | 2.44 | 1.66 | | 17.6 | 29.1 | | 101.8 | 50.2 | | 1912 | 1964 | | 0.33 | 0.62 | | 0.37 | 1.03 |
| 42-44 | | 20.2 | 56.6 | | | | | 8.2 | 30.6 | | 2.36 | 1.15 | | 15.1 | 39.6 | | 110.5 | 52.7 | | 1904 | 1961 | | 0.32 | 0.62 | | 0.34 | 0.95 |
| 44-46 | | | 56.5 | | | | | | 30.5 | | | 1.04 | | | 36.8 | | | 56.4 | | | 1958 | | | 0.60 | | | 0.99 |
| 46-48 | | | 53.5 | | | | | | 27.5 | | | 2.22 | | | 66.0 | | | 60.3 | | | 1954 | | | 0.59 | | | 0.49 |
| 48-50 | | | 49.7 | | | | | | 23.7 | | | 2.00 | | | 46.8 | | | 68.9 | | | 1945 | | | 0.54 | | | 0.55 |
| 50-52 | | | 43.4 | | | | | | 17.4 | | | 2.55 | | | 40.9 | | | 76.6 | | | 1937 | | | 0.51 | | | 0.43 |
| 52-54 | | | 40.7 | | | | | | 14.7 | | | 2.11 | | | 25.7 | | | 85.4 | | | 1929 | | | 0.48 | | | 0.54 |
| 54-56 | | | 35.7 | | | | | | 9.7 | | | 2.59 | | | 20.7 | | | 92.5 | | | 1922 | | | 0.46 | | | 0.43 |
| 56-58 | | | 38 | | | | | | 12 | | | 1.61 | | | 18.7 | | | 99.5 | | | 1914 | | | 0.44 | | | 0.57 |
| 58-60 | | | 35.2 | | | | | | 9.2 | | | 2.21 | | | 19.1 | | | 107.6 | | | 1906 | | | 0.42 | | | 0.41 |




Appendix A: Numbers, relative abundance (%), and statistical data for benthic foraminifera from sediment core 11257, taken from western area of Gamak Bay.

| Species / Depth(cm) | 0-2 | 2-4 | 4-6 | 6-8 | 8-10 | 10-12 | 12-14 | 14-16 | 16-18 | 18-20 | 20-22 | 22-24 | 24-26 | 26-28 | 28-30 | 30-32 | 32-34 | 34-36 | 36-38 | 38-40 |
|---|---|---|---|---|---|---|---|---|---|---|---|---|---|---|---|---|---|---|---|---|
| **Agglutinated Foram** | | | | | | | | | | | | | | | | | | | | |
| Ammobaculites agglutinans | 1.4 | 0.7 | 1.4 | | | 0.4 | 1.3 | 0.7 | 0.2 | 0.5 | 0.3 | | 0.2 | 0.3 | | | | | | 0.6 |
| Ammobaculites cubensis | 6.1 | 6.9 | 2.8 | 5.9 | 6.0 | 6.7 | 5.0 | 4.8 | 4.7 | 4.3 | 5.0 | 4.9 | 4.8 | 6.3 | 3.0 | | 5.2 | 3.0 | | 4.8 |
| Eggerella advena | 1.4 | 3.8 | | 1.2 | 1.7 | | | 0.3 | 0.2 | | | | | | | | | | | 0.4 |
| Trochammina hadai | 4.2 | 2.4 | 4.2 | 6.7 | 2.1 | 2.2 | 4.7 | 3.8 | 0.9 | 5.4 | 1.9 | 2.1 | 1.2 | | 1.9 | | 2.4 | 0.7 | | 0.6 |
| **C.-H. Foram** | | | | | | | | | | | | | | | | | | | | |
| Ammonia beccarii | 20.8 | 11.1 | 19.3 | 13.7 | 15.8 | 16.0 | 11.7 | 11.0 | 14.0 | 12.1 | 15.3 | 11.3 | 14.4 | 11.3 | 14.4 | 13.9 | 13.4 | 12.6 | 12.5 | 15.0 |
| Ammonia ketienziensis | 12.3 | 6.9 | 4.2 | 6.3 | 6.4 | 6.7 | 5.7 | 9.0 | 7.4 | 6.7 | 5.0 | 6.4 | 6.7 | 5.8 | 2.7 | 8.8 | 6.5 | 7.2 | 5.7 | 3.8 |
| Anomalina sp. | | | | | | | | | | 0.3 | | | | | | | | | | |
| Astrononion sp. | 0.5 | 0.7 | | 0.4 | 0.9 | 0.7 | | 0.7 | | 0.8 | | 1.0 | 0.5 | 2.1 | 1.5 | 1.4 | 1.1 | 0.5 | 1.5 | 0.6 |
| Bolivina robusta | 0.9 | 2.1 | 0.8 | 0.4 | 0.9 | | 0.7 | 1.0 | 0.7 | 1.1 | 2.5 | 0.5 | 1.0 | 1.6 | 0.8 | 1.0 | 0.2 | 1.2 | 1.8 | 1.2 |
| Bolivina striatula | 0.9 | 1.0 | 1.7 | | 0.9 | 1.1 | 1.3 | 1.4 | 2.3 | 1.1 | 1.9 | 1.8 | 2.2 | 1.0 | 2.3 | 1.7 | 1.7 | 1.9 | 1.8 | 1.4 |
| Bolivina subexcavata | | | 0.8 | | 0.4 | 1.1 | 1.7 | | 0.5 | 0.5 | | 0.5 | 0.7 | 0.3 | | 0.7 | 0.4 | 1.2 | 0.7 | 0.6 |
| Bolivina sp. | | | | | | 0.7 | | 0.3 | | | 0.3 | | | 0.3 | | 0.3 | | | | |
| Buccella frigida | 3.8 | 4.2 | 3.7 | 3.9 | 3.8 | 3.0 | 4.0 | 3.4 | 4.0 | 7.5 | 5.0 | 3.9 | 5.3 | 4.7 | 4.6 | 7.4 | 3.5 | 7.0 | 4.6 | 5.0 |
| Bullimina exilis | | | | | | | 0.7 | | | | | | | | | | | | | |
| Bulimina marginata | | 0.3 | 0.3 | 0.8 | 0.9 | | 0.7 | 0.3 | 0.5 | | 0.3 | 1.3 | 1.0 | 0.5 | 0.8 | | 0.6 | 0.9 | 0.2 | 0.4 |
| Buliminella sp. | 0.5 | | | | | | | | | | 0.3 | | | 0.3 | | 0.3 | 0.4 | | 0.2 | 0.2 |
| Cancris auriculus | | | | | | | 0.7 | | | | 0.3 | | | | | | 0.3 | 0.7 | | 0.2 |
| Cibicides lobatulus | 0.5 | 1.7 | 1.1 | 2.0 | 1.3 | 1.5 | | 1.0 | 1.4 | 1.6 | 0.9 | 1.0 | 1.0 | 1.3 | 1.5 | 1.0 | 0.4 | 1.9 | 1.5 | 1.4 |
| Dentalina sp. | | | | | 0.4 | | | | | | | | | | | | | | | |
| Eilohedra nipponica | 0.9 | 1.4 | 0.6 | 0.4 | 1.3 | 1.1 | 1.0 | 0.3 | | 1.6 | 0.6 | 1.3 | | 1.0 | 1.1 | 0.3 | 1.1 | 1.2 | 2.0 | 1.0 |
| Elphidium advenum | 10.8 | 16.0 | 12.5 | 17.3 | 12.8 | 19.0 | 13.0 | 13.8 | 12.1 | 11.0 | 15.3 | 12.6 | 12.5 | 16.3 | 13.3 | 12.5 | 12.1 | 10.7 | 11.0 | 12.0 |
| Elphidium clavatum | 11.3 | 14.2 | 9.1 | 8.2 | 10.3 | 13.8 | 11.3 | 11.7 | 14.0 | 10.2 | 12.5 | 15.2 | 16.8 | 14.2 | 16.0 | 16.6 | 15.2 | 15.2 | 10.7 | 16.0 |
| Elphidium somaense | 2.4 | 2.1 | 6.2 | 3.5 | 5.6 | 3.7 | 5.7 | 5.9 | 5.6 | 4.6 | 2.8 | 5.4 | 3.4 | 3.9 | 5.3 | 4.4 | 6.3 | 5.1 | 6.8 | 5.6 |
| Elphidium subarcticum | 10.4 | 13.9 | 14.7 | 12.5 | 13.7 | 9.0 | 14.0 | 10.3 | 13.7 | 9.7 | 13.7 | 13.6 | 8.6 | 10.0 | 13.3 | 14.5 | 10.6 | 11.9 | 19.3 | 11.0 |
| Elphidium subincertum | 0.5 | | 0.8 | 0.4 | 0.4 | 1.1 | 0.3 | 1.0 | 0.5 | 1.6 | 0.9 | 0.5 | 1.4 | 1.3 | 1.1 | 2.0 | 0.4 | 1.4 | 1.3 | 0.2 |
| Favulina hexagona | | | | | | | | | | | | | | 0.3 | | | | | | |
| Fissurina laevigata | | | 0.3 | | 0.9 | 1.1 | 0.7 | | 0.2 | 0.8 | 0.6 | 0.5 | 1.0 | 0.3 | | 0.3 | 0.4 | 0.2 | 0.7 | |
| Fissurina marginata | 0.9 | | | | | | 1.0 | | | | 0.9 | 0.8 | 0.2 | 0.3 | | 0.3 | 0.4 | | 0.2 | 0.2 |
| Globocassidulina subglobosa | | | | | 0.4 | | 0.7 | | 0.2 | 0.3 | | | | 0.5 | 0.4 | | | | | 0.4 |
| Gyroidinoides cushmani | | | 0.6 | | | | | 0.3 | | 0.3 | | | 0.5 | | | | | | | |
| Gyroidinoides nipponicus | | | | | | | | | | | | | | 0.3 | | | | | | |
| Hanzawaia sp. | 0.5 | | 0.3 | | | | | | | | | | 0.5 | | | | 0.2 | 0.5 | 0.4 | |
| Hyalinea sp. | | | 0.3 | | | | | | | 0.3 | | | | | | | | | | |
| Islandiella japonica | | 0.3 | 0.3 | 0.4 | 0.4 | 0.4 | | | | | | | | 0.8 | | 0.3 | 0.2 | | 0.4 | |
| Melonis barleeanus | 0.9 | 0.3 | 1.4 | 1.2 | | | 1.3 | 0.3 | 0.9 | 0.8 | | | 0.5 | 0.3 | 0.8 | 1.0 | 0.2 | 1.2 | 0.4 | |
| Murreynella sp. | | 0.7 | | 0.4 | 0.4 | | | 0.3 | 0.7 | | | 0.3 | 0.2 | | 0.8 | | 0.4 | | 0.2 | 0.4 |
| Nonionella globosa | | | 0.8 | | | | 1.0 | | 0.5 | | | 0.5 | | | | 0.3 | 0.2 | | 0.4 | |
| Paracassidulina neocarinata | | | | | | | | | | | | 0.3 | 0.7 | | 0.8 | | 0.4 | 0.5 | 0.7 | 1.4 |
| Paracassidulina sagaminiensis | | | 1.4 | | | 1.1 | 0.7 | | 0.7 | 1.9 | 0.3 | 0.5 | | 0.3 | | | | | | |
| Pararotalia nipponica | | | | | | | | | | | | 0.5 | | | | | 3.0 | | 3.5 | |
| Pararotalia sp. | | 1.0 | 0.8 | 2.0 | 0.9 | 0.4 | 0.7 | 3.1 | 1.4 | 5.1 | 0.6 | 5.4 | 3.4 | 6.0 | 3.4 | 1.4 | | 4.2 | | 3.4 |
| Pseudoeponides japonicus | 0.5 | | 0.3 | | 0.4 | | 0.3 | 0.3 | 0.5 | | | 0.3 | 1.0 | | 0.4 | 0.7 | 0.2 | | 0.4 | 0.2 |
| Pseudononion japonicum | 0.5 | 0.7 | | | 0.4 | | 0.3 | 0.7 | | | 0.0 | 1.0 | 0.5 | | 0.7 | 0.9 | 0.5 | | | |
| Pseudoparrella naraensis | 2.8 | 3.8 | 4.2 | 5.5 | 3.4 | 4.5 | 4.3 | 4.8 | 3.3 | 3.8 | 3.1 | 1.8 | 3.8 | 3.1 | 1.9 | 1.7 | 4.3 | 3.0 | 3.7 | 3.8 |
| Pseudoparrella tamana | 0.9 | | 1.7 | 1.6 | 3.0 | | 0.3 | 1.0 | 1.9 | | 1.6 | 1.5 | 1.2 | 1.3 | 1.1 | 1.0 | 1.1 | 0.9 | 0.4 | 0.8 |
| Pseudorotalia gaimardii | 0.5 | 0.3 | 0.3 | | 0.9 | 1.1 | | | 0.2 | 0.5 | 0.6 | | 0.2 | | 0.8 | 0.3 | 0.4 | | 1.1 | |
| Pullenia quinqueloba | 0.5 | 0.7 | | 0.4 | 0.4 | | | | 0.5 | 0.3 | | | 0.2 | | | | | | 0.4 | 0.2 |
| Rectobolivina raphana | | | 0.3 | | | | | | | | | 0.3 | | 0.5 | | | | | | |
| Rosalina bradyi | | | | | | | | | 0.5 | | | | 0.2 | | | | | 0.2 | | |
| Rosalina globularis | | | 0.6 | 0.8 | | | 0.7 | 1.4 | 0.7 | | 0.9 | 0.8 | 0.2 | 0.5 | 1.1 | 0.3 | 0.6 | 0.5 | 0.4 | 0.6 |
| Rosalina vilardeboana | | | | | | | 1.0 | | | 0.8 | | | | | | | | | | 1.0 |
| Rosalina sp. | | 0.3 | | 0.4 | | | 0.7 | 0.3 | 0.2 | | | | | | 0.4 | | 0.2 | | | |
| Reussella sp. | 0.5 | 0.3 | | | 0.4 | | | | 0.2 | | | | | | 0.4 | 0.3 | | | | |
| Lagena gracillima | | | | | | | | | | 0.3 | | | | | | | 0.2 | | | |
| Lagena perlucida | | | | | | | | 0.3 | | | | | | | 0.8 | | | | | |
| Lagena sulcata spicata | | | | | | 0.4 | | | | | | | 0.3 | | | | | 0.2 | 0.2 | |
| Uvigerina nitidula | | | | | | | | 0.3 | | | | | | | | | | | | |
| Uvigerina proboscidea | | | | | | | | 0.3 | 0.3 | | | | 0.7 | | | | | 0.2 | | |
| Uvigerinella glabra | | | 0.6 | 0.8 | | | | 0.7 | | 0.5 | 0.3 | | | | | 0.3 | 0.6 | | 0.2 | |
| Valvulineria sadonica | 0.9 | | 0.6 | | | | 1.7 | 0.7 | 0.2 | | 1.2 | 0.3 | 0.5 | | | | | | | 0.6 |
| **C.-P. Foram** | | | | | | | | | | | | | | | | | | | | |
| Quinqueloculina contorta | | 1.7 | | 1.2 | 0.9 | 0.7 | | 1.0 | 0.2 | | 0.9 | 0.3 | 0.5 | 0.5 | 0.8 | 1.0 | 1.3 | 0.7 | 0.7 | |
| Quinqueloculina lamarckiana | 0.5 | | | | | | | 0.3 | 0.9 | 0.3 | 0.3 | 0.5 | | 0.3 | | 0.7 | | 0.2 | | 1.6 |
| Quinqueloculina seminula | 0.5 | | 0.8 | 1.6 | 2.1 | 1.5 | 2.0 | 1.7 | 2.3 | 2.9 | 2.8 | 1.3 | 1.4 | 1.3 | 2.3 | 0.7 | 1.5 | 2.8 | 2.9 | 2.8 |
| Sigmoilopsis sp. | | | | 0.4 | | 0.4 | | 0.3 | | 0.3 | 0.9 | | 0.5 | | | 0.7 | 0.9 | | 0.4 | |
| Spiroloculina sp. | | | | | | | | | | 0.5 | | | | 0.5 | | 0.4 | | | | |
| Miliolinella sp. | | | 0.3 | | | | | | | 0.5 | | 0.8 | | | | | | | | 0.2 |
| Total No. of Benthic Foram | 212 | 288 | 353 | 255 | 234 | 268 | 300 | 290 | 430 | 373 | 321 | 389 | 417 | 381 | 263 | 296 | 462 | 429 | 456 | 501 |
| Species Number S | 31 | 28 | 35 | 29 | 32 | 28 | 34 | 38 | 39 | 35 | 33 | 36 | 38 | 37 | 32 | 36 | 40 | 33 | 37 | 37 |
| Percentage of A. Foram | 13.2 | 13.9 | 8.5 | 13.7 | 9.8 | 9.3 | 11.0 | 9.7 | 6.0 | 10.2 | 7.2 | 6.9 | 6.2 | 6.6 | 4.9 | | 7.6 | 3.7 | | 6.4 |
| Percentage of C.-H. Foram | 85.8 | 84.4 | 90.4 | 83.1 | 87.2 | 88.1 | 87.0 | 86.9 | 90.5 | 85.3 | 87.9 | 90.2 | 91.4 | 90.8 | 91.6 | 96.3 | 88.3 | 92.5 | 95.6 | 88.4 |
| Percentage of C.-P. Foram | 0.9 | 1.7 | 1.1 | 3.1 | 3.0 | 2.6 | 2.0 | 3.4 | 3.5 | 4.6 | 5.0 | 2.8 | 2.4 | 2.6 | 3.4 | 3.7 | 4.1 | 3.7 | 4.4 | 5.2 |
| Total No. of Planktic Foram | 3 | 16 | 17 | 13 | 9 | 5 | 21 | | 31 | | 5 | 28 | 20 | 27 | 19 | 10 | 26 | 23 | 38 | 33 |
| Species Diversity H(s) | 2.66 | 2.68 | 2.77 | 2.72 | 2.79 | 2.63 | 2.88 | 2.91 | 2.81 | 2.94 | 2.76 | 2.82 | 2.85 | 2.81 | 2.78 | 2.69 | 2.86 | 2.82 | 2.79 | 2.81 |
| Equitability E | 0.77 | 0.80 | 0.78 | 0.81 | 0.81 | 0.79 | 0.82 | 0.80 | 0.77 | 0.83 | 0.79 | 0.79 | 0.78 | 0.78 | 0.80 | 0.75 | 0.78 | 0.81 | 0.77 | 0.78 |
| Total No. of B. F. in 20ml of sd. | 6784 | 9216 | 11296 | 8160 | 7488 | 4288 | 4800 | 4640 | 6880 | 5968 | 5136 | 6224 | 6672 | 6096 | 4208 | 4736 | 7392 | 6864 | 7296 | 8016 |
| Total No. of P. F. in 20ml of sd. | 96 | 512 | 544 | 416 | 288 | 80 | 336 | | 496 | | 80 | 448 | 320 | 432 | 304 | 160 | 416 | 368 | 608 | 528 |

A.=Agglutinated, C.-H.= Calcareous-Hyaline, C.-P.=Calcareous-Porcelaneous, B. F.=Benthic Foraminifera, P. F.=Planktic Foraminifera, sd.=sediments





Appendix B: Numbers, relative abundance (%), and statistical data for benthic foraminifera from sediment core 10863, taken from eastern area of Gamak Bay.

| Species / Depth(cm) | 0-2 | 2-4 | 4-6 | 6-8 | 8-10 | 10-12 | 12-14 | 14-16 | 16-18 | 18-20 | 20-22 | 22-24 | 24-26 | 26-28 | 28-30 | 30-32 | 32-34 | 34-36 | 36-38 | 38-40 | 40-42 | 42-44 |
|---|---|---|---|---|---|---|---|---|---|---|---|---|---|---|---|---|---|---|---|---|---|---|
| **Agglutinated Foram** | | | | | | | | | | | | | | | | | | | | | | |
| *Ammobaculites agglutinans* | 0.4 | 0.5 | | | | | | | | | | | | | 0.6 | | | | | | | |
| *Ammobaculites cubensis* | 1.7 | 2.9 | | 0.4 | 1.3 | 1.4 | 1.0 | 0.5 | 0.8 | | 5.9 | 6.5 | 3.5 | 3.8 | 4.5 | 1.1 | 0.3 | 3.4 | 0.2 | 1.3 | 6.9 | 0.6 |
| *Eggerella advena* | 1.7 | 1.0 | | | 0.5 | 0.4 | | | | | | 1.3 | 0.3 | 0.8 | | | | 0.3 | | 0.2 | | |
| *Trochammina hadai* | 3.3 | 1.5 | 2.8 | 0.8 | 1.5 | 0.7 | 0.3 | | 0.4 | | 4.3 | 3.0 | 3.7 | 2.5 | 1.8 | | | 0.3 | 0.5 | 0.7 | 0.7 | |
| **C.-H. Foram** | | | | | | | | | | | | | | | | | | | | | | |
| *Ammonia beccarii* | 9.6 | 14.1 | 11.7 | 19.3 | 13.5 | 15.4 | 22.0 | 19.1 | 14.6 | 18.8 | 14.6 | 15.2 | 14.4 | 13.5 | 15.9 | 18.8 | 17.1 | 13.7 | 14.6 | 11.7 | 10.7 | 10.1 |
| *Ammonia ketienziensis* | 7.5 | 8.7 | 4.4 | 8.0 | 6.3 | 8.8 | 8.7 | 6.7 | 9.6 | 12.7 | 7.6 | 7.8 | 13.1 | 11.0 | 10.8 | 13.6 | 12.5 | 6.7 | 9.6 | 8.0 | 9.3 | 10.5 |
| *Astrononion* sp. | 0.4 | | | | | 1.1 | | 0.5 | | | | | 0.5 | | | 0.6 | 0.3 | 0.8 | | 1.2 | 2.4 | 1.0 |
| *Bolivina robusta* | 1.7 | 0.5 | 1.7 | 0.8 | | 1.1 | 1.0 | 1.0 | 0.4 | 1.5 | 0.5 | 0.9 | 1.3 | 0.8 | 0.6 | 1.1 | 0.3 | 0.8 | 1.2 | 1.4 | 1.7 | 0.8 |
| *Bolivina striatula* | | | 1.1 | | 0.5 | 1.4 | 0.3 | 1.0 | 0.8 | | 0.5 | 0.9 | 0.5 | 0.3 | | 2.8 | 0.3 | 1.8 | 1.7 | 0.6 | 1.4 | 1.0 |
| *Bolivina subexcavata* | | | | | | | | | | | | | 0.3 | 0.3 | | | | | | 0.1 | | 0.2 |
| *Buccella frigida* | 4.2 | 1.9 | 2.8 | 4.4 | 4.1 | 4.9 | 4.2 | 4.1 | 4.2 | 5.1 | 3.2 | 4.8 | 2.7 | 4.4 | 1.8 | 3.4 | 4.9 | 3.1 | 3.6 | 3.1 | 5.5 | 3.1 |
| *Bulimina marginata* | 0.8 | | 0.6 | 0.8 | 0.8 | 0.4 | 0.3 | | | 0.5 | | | | 0.8 | | 1.1 | 1.0 | | | 0.4 | 0.3 | 0.8 |
| *Buliminella* sp. | | | | | | | | | | | | | | | | | | | | 0.2 | 0.3 | 0.2 |
| *Cancris auriculus* | | | | | | | | | | | | | | | | | | | | 0.1 | | |
| *Cibicides lobatulus* | | | | 1.6 | 1.0 | | 1.4 | 2.1 | | 0.5 | 1.6 | | 0.5 | 1.6 | 1.5 | | 2.0 | 2.3 | 2.2 | 1.6 | 0.7 | 1.3 |
| *Eilohedra nipponica* | 0.8 | | | | | | | 0.5 | 0.8 | | 0.5 | | 1.1 | 1.1 | 0.9 | 0.6 | 1.0 | 1.3 | 0.5 | 1.5 | 1.4 | 1.7 |
| *Elphidium advenum* | 7.9 | 7.3 | 7.8 | 8.4 | 5.8 | 8.1 | 12.9 | 6.2 | 10.0 | 13.2 | 12.4 | 11.7 | 13.1 | 13.7 | 13.0 | 20.5 | 16.4 | 13.0 | 13.7 | 11.5 | 17.9 | 17.2 |
| *Elphidium clavatum* | 9.2 | 9.7 | 10.0 | 4.8 | 8.6 | 11.2 | 10.8 | 12.9 | 12.3 | 12.2 | 13.5 | 12.2 | 12.8 | 13.2 | 11.7 | 14.2 | 19.1 | 14.8 | 14.4 | 19.0 | 14.8 | 13.0 |
| *Elphidium somaense* | 5.0 | 6.3 | 8.3 | 4.4 | 5.1 | 5.6 | 3.1 | 0.5 | 7.3 | 7.6 | 5.9 | 7.0 | 5.6 | 7.4 | 3.6 | 3.4 | 2.0 | 4.1 | 6.5 | 6.2 | 2.7 | 6.5 |
| *Elphidium subarcticum* | 40.8 | 35.0 | 36.7 | 37.8 | 42.6 | 30.2 | 28.6 | 34.0 | 33.0 | 14.7 | 14.1 | 12.6 | 7.5 | 9.6 | 11.7 | 8.5 | 8.9 | 12.4 | 11.5 | 11.4 | 6.2 | 12.6 |
| *Elphidium subincertum* | | | 1.1 | 0.8 | 0.5 | 0.7 | 0.7 | 0.5 | 0.8 | 0.5 | | 0.4 | 0.5 | 0.5 | 0.6 | 0.6 | 1.0 | 1.3 | 1.0 | 0.6 | 1.4 | 0.8 |
| *Favulina hexagona* | | | | | | | | | | | | | | | | 0.6 | | | | | | |
| *Fissurina laevigata* | | | | 0.8 | 0.3 | 0.7 | | | | | 1.1 | | 1.1 | | | 0.3 | 0.6 | | | | 0.3 | |
| *Fissurina marginata* | | 0.5 | | | | | | | | 1.0 | | | 0.3 | | | | | | 0.2 | 0.4 | | 0.2 |
| *Globocassidulina subglobosa* | | | | | | | | 0.3 | | | | | 0.3 | | | 0.6 | | 0.3 | | 0.5 | 0.4 | 0.8 |
| *Gyroidinoides cushmani* | | | | | 0.3 | | | | | | | | 0.5 | | | | | | | 0.2 | | |
| *Gyroidinoides nipponicus* | | | | | 0.3 | | | | | | | | | 0.3 | | | | | | | 0.1 | 0.3 |
| *Hanzawaia* sp. | | | | 0.4 | | | | | | | 1.6 | | 0.8 | 0.5 | | | | | | 0.2 | 0.2 | 0.3 |
| *Islandiella japonica* | 0.4 | | | | | | | | | | | | 0.4 | | | 0.6 | | 0.3 | | | | |
| *Melonis barleeanus* | | 1.0 | 0.6 | | 0.3 | 0.4 | | | | 1.5 | | | | 0.8 | 1.2 | 1.1 | | 0.3 | 1.7 | 0.2 | | 0.8 |
| *Murreynella* sp. | | 0.5 | 0.6 | 0.4 | | | | 1.0 | 0.4 | 0.5 | 0.5 | 0.9 | | | | | 1.3 | | 0.2 | | | |
| *Nonionella globosa* | | | | | | 0.4 | | | 0.4 | | | | 1.1 | | 0.9 | | | 0.5 | | | | |
| *Paracassidulina neocarinata* | | | 2.2 | | | | | 2.1 | | | | | 0.8 | | 0.3 | | | | 0.2 | 0.2 | | 0.8 |
| *Paracassidulina sagaminiensis* | | 0.5 | | | 0.5 | 1.1 | | | | | | 1.3 | 0.5 | 0.3 | 0.9 | | | | 0.7 | 0.5 | | |
| *Pararotalia nipponica* | | | | 0.4 | 0.3 | | | | | | | | 0.3 | | | 0.6 | | | | 0.2 | | |
| *Pararotalia* sp. | 1.3 | 1.0 | 2.2 | | 0.3 | | 0.7 | 2.1 | 1.5 | 0.5 | | 2.2 | 2.4 | 1.6 | 4.5 | | 1.3 | 3.6 | 2.9 | 2.8 | | 3.6 |
| *Pseudoeponides japonicus* | | | | | | | | | | | | | 0.4 | 0.5 | | | | 0.3 | 0.2 | 0.1 | 0.3 | |
| *Pseudononion japonicum* | | | | | | | | | | 0.5 | | | 0.5 | 0.5 | | | 0.3 | 1.0 | 0.2 | | | 0.4 |
| *Pseudoparrella naraensis* | 1.3 | 1.9 | 1.7 | 2.4 | 2.0 | 1.8 | 0.7 | 3.1 | 1.5 | 4.6 | 4.3 | 2.6 | 3.5 | 3.0 | 3.0 | 0.6 | 2.6 | 4.1 | 4.6 | 3.3 | 1.0 | 2.9 |
| *Pseudoparrella tamana* | | 1.9 | 2.2 | 2.4 | 2.0 | 2.1 | 2.1 | 2.1 | 0.8 | 2.0 | 2.2 | 2.2 | 1.6 | 1.1 | 1.5 | 1.7 | 2.3 | 1.0 | 1.9 | 1.6 | 3.8 | 1.7 |
| *Pseudorotalia gaimardii* | 0.4 | | | 0.8 | | | 0.7 | | | 0.5 | 0.5 | 0.4 | 0.8 | 1.1 | 1.2 | | 1.0 | | 0.2 | 0.7 | 1.4 | 0.8 |
| *Pullenia quinqueloba* | | | | | | | | | | | | | | | 0.3 | | | 0.3 | | | | |
| *Rectobolivina raphana* | | | | | | | | | | | | | | | 0.3 | | | 0.2 | | | | |
| *Rosalina bradyi* | | | | | 0.3 | | | | | | | 0.9 | 0.5 | 0.8 | | 0.6 | 1.0 | | | 0.4 | 0.7 | 0.6 |
| *Rosalina globularis* | 0.8 | 0.5 | 0.6 | | 0.3 | | | | 0.4 | | | 1.3 | | 0.5 | | 1.1 | 0.7 | 0.8 | 0.7 | 0.6 | | 1.0 |
| *Rosalina* sp. | 0.4 | 0.5 | | | | | | | | | | | | | | | | | | | | |
| *Lagena sulcata spicata* | | 1.0 | | | | | | | | | | | | | | | | | | 0.1 | | |
| *Lenticulina* sp. | | 0.5 | | | | | | | | | | | | | | | | | | | | |
| *Uvigerina proboscidea* | | 0.5 | | | | | | | | | | 0.4 | | | | | 0.6 | | | | | |
| *Valvulineria sadonica* | | | | | 0.3 | 0.7 | | | 0.4 | 1.0 | | | | | 0.9 | | | 0.8 | 0.2 | 0.5 | | 0.6 |
| **C.-P. Foram** | | | | | | | | | | | | | | | | | | | | | | |
| *Quinqueloculina contorta* | | | | | | | | | | 0.5 | | | 0.3 | 1.1 | 0.3 | | 0.3 | 1.0 | 0.7 | 0.8 | 1.4 | 0.6 |
| *Quinqueloculina lamarckiana* | | | | | | | | | | | | | 0.8 | | 0.9 | 1.1 | 1.3 | 1.6 | 0.5 | 0.7 | 2.1 | 1.0 |
| *Quinqueloculina seminula* | 0.4 | 0.5 | 1.1 | | 1.0 | 1.8 | | | | | 3.8 | 2.6 | 2.1 | 2.2 | 3.3 | 1.1 | | 2.6 | 2.2 | 4.4 | 3.4 | 2.1 |
| *Sigmoilopsis* sp. | | | | | | | | | | | 0.5 | | | | 0.6 | | 0.3 | 0.5 | | | 0.3 | 0.2 |
| *Spiroloculina* sp. | | | | | | | | | | | 0.5 | | 0.3 | | 0.3 | | | 0.5 | 0.2 | 0.3 | 0.3 | |
| *Miliolinella* sp. | | | | | | | | | | | | | 0.3 | | | | | 0.8 | | 0.2 | | |
| Total No. of Benthic Foram | 240 | 206 | 180 | 249 | 394 | 285 | 287 | 194 | 261 | 197 | 185 | 230 | 375 | 364 | 333 | 176 | 304 | 386 | 417 | 1008 | 291 | 477 |
| Species Number S | 22 | 25 | 20 | 20 | 27 | 23 | 19 | 19 | 20 | 21 | 22 | 25 | 37 | 31 | 33 | 25 | 27 | 33 | 35 | 43 | 30 | 33 |
| Percentage of A. Foram | 7.1 | 5.8 | 2.8 | 1.2 | 3.3 | 2.5 | 1.4 | 0.5 | 1.1 | | 10.3 | 10.9 | 7.5 | 7.1 | 6.9 | 1.1 | 0.3 | 3.9 | 0.7 | 2.2 | 7.6 | 1 |
| Percentage of C.-H. Foram | 92.5 | 93.7 | 96.1 | 98.8 | 95.7 | 95.8 | 98.6 | 99.5 | 98.9 | 99.5 | 84.9 | 86.5 | 88.8 | 89.6 | 87.7 | 96.6 | 97.7 | 89.1 | 95.7 | 91.5 | 84.9 | 95 |
| Percentage of C.-P. Foram | 0.4 | 0.5 | 1.1 | | 1.0 | 1.8 | | | | 0.5 | 4.9 | 6.4 | 3.7 | 3.3 | 5.4 | 2.3 | 2.0 | 7.0 | 3.6 | 6.3 | 7.6 | 4 |
| Total No. of Planktic Foram | 5 | 14 | 9 | 1 | 10 | | 6 | 2 | 3 | 5 | | 13 | 20 | 23 | 24 | | 10 | 24 | 27 | 91 | | |
| Species Diversity H(s) | 2.17 | 2.28 | 2.25 | 2.09 | 2.12 | 2.33 | 2.10 | 2.14 | 2.12 | 2.38 | 2.59 | 2.68 | 2.83 | 2.75 | 2.79 | 2.42 | 2.46 | 2.80 | 2.70 | 2.80 | 2.74 | 2.75 |
| Equitability E | 0.70 | 0.71 | 0.75 | 0.70 | 0.64 | 0.74 | 0.71 | 0.73 | 0.71 | 0.78 | 0.84 | 0.83 | 0.78 | 0.80 | 0.80 | 0.75 | 0.75 | 0.80 | 0.76 | 0.74 | 0.81 | 0.79 |
| Total No. of B. F. in 20ml of sd. | 1920 | 1648 | 2880 | 996 | 1576 | 2280 | 1148 | 1552 | 2088 | 1576 | 2960 | 3680 | 3000 | 2912 | 5328 | 1408 | 2432 | 3088 | 3336 | 4032 | 2328 | 3816 |
| Total No. of P. F. in 20ml of sd. | 40 | 112 | 144 | 4 | 40 | | 24 | 16 | 24 | 40 | | 208 | 160 | 184 | 384 | | 80 | 192 | 216 | 364 | | |

A.=Agglutinated, C.-H.= Calcareous-Hyaline, C.-P.=Calcareous-Porcelaneous, B. F.=Benthic Foraminifera, P. F.=Planktic Foraminifera, sd.=sediments





Appendix C: Numbers, relative abundance (%), and statistical data for benthic foraminifera from sediment core 11285, taken from northwestern area of Gamak Bay.

| Species Depth(cm) | 0-2 | 2-4 | 4-6 | 6-8 | 8-10 | 10-12 | 12-14 | 14-16 | 16-18 | 18-20 | 20-22 | 22-24 | 24-26 | 26-28 | 28-30 | 30-32 | 32-34 | 34-36 | 36-38 | 38-40 | 40-42 | 42-44 | 44-46 | 46-48 | 48-50 | 50-52 | 52-54 | 54-56 | 56-58 | 58-60 |
|---|---|---|---|---|---|---|---|---|---|---|---|---|---|---|---|---|---|---|---|---|---|---|---|---|---|---|---|---|---|---|
| **Agglutinated Foram** | | | | | | | | | | | | | | | | | | | | | | | | | | | | | | |
| *Ammobaculites agglutinans* | 1.2 | 0.9 | 0.3 | 0.2 |  | 1.1 |  |  |  |  |  |  |  |  | 0.7 |  |  | 1.4 | 0.7 | 0.3 |  |  |  |  |  |  |  |  |  |  |
| *Ammobaculites cubensis* |  | 0.4 |  | 0.4 | 0.3 | 1.1 |  | 1.8 |  |  |  |  |  |  | 0.7 | 3.0 | 2.1 | 2.1 |  | 1.6 | 1.2 |  |  |  | 0.1 |  |  |  |  | 2.1 |
| *Eggerella advena* | 8.2 | 4.7 | 3.7 | 13.8 | 4.2 | 4.5 | 4.0 |  | 0.6 | 3.3 |  |  |  | 1.9 | 2.7 | 4.5 | 1.1 | 2.7 |  | 0.3 |  |  |  |  |  |  |  |  |  |  |
| *Textularia mariae* |  |  |  |  |  |  |  |  |  |  |  |  |  |  |  |  | 0.7 |  |  |  |  |  |  |  |  |  |  |  |  |  |
| *Trochammina hadai* | 18.5 | 18.1 | 4.9 | 12.2 | 4.2 | 17.0 | 17.2 | 10.9 | 1.9 | 7.9 | 18.9 | 1.1 | 7.2 | 11.5 | 4.7 | 45.5 | 21.1 | 0.7 | 9.7 | 11.3 | 1.8 | 1.0 |  |  |  |  |  |  | 0.4 |  |
| **C.-H. Foram** | | | | | | | | | | | | | | | | | | | | | | | | | | | | | | |
| *Ammonia beccarii* | 14.0 | 12.6 | 15.0 | 8.1 | 8.2 | 33.0 | 37.4 | 30.0 | 18.1 | 13.2 | 3.3 | 9.7 | 8.0 | 11.5 | 32.2 | 9.1 | 9.5 | 17.1 | 17.5 | 23.6 | 17.9 | 45.1 | 84.2 | 68.7 | 55.2 | 44.6 | 49.6 | 41.8 | 37.1 | 48.3 |
| *Bolivina* sp. |  |  |  |  | 0.2 |  |  |  | 0.3 |  | 0.5 |  |  |  |  |  | 1.1 | 2.7 |  |  | 0.6 |  |  |  |  |  |  |  |  |  |
| *Buccella frigida* | 2.3 | 2.8 | 2.6 | 4.3 | 6.1 | 6.8 | 10.1 | 4.5 | 10.6 | 4.6 | 8.5 | 40.1 | 18.8 | 25.0 | 12.1 | 10.6 | 27.4 | 21.9 | 58.7 | 14.9 | 25.0 | 42.2 | 12.3 | 6.0 | 10.9 | 6.8 | 14.0 | 18.4 | 27.3 | 11.0 |
| *Bulimina marginata* |  |  |  |  |  |  |  |  |  |  |  |  |  |  |  |  |  | 1.4 |  |  |  |  |  |  |  |  |  |  |  |  |
| *Cancris auriculus* |  |  |  |  |  |  |  |  |  |  |  |  |  |  |  |  | 0.7 |  |  |  |  |  |  |  |  |  |  |  |  |  |
| *Cibicides lobatulus* |  |  |  |  |  | 1.1 |  |  |  |  |  |  |  |  |  |  | 1.5 | 2.1 |  |  |  |  |  |  |  |  |  |  |  |  |
| *Eilohedra nipponica* |  |  |  |  |  |  |  |  |  |  |  |  |  |  |  |  |  | 3.4 |  |  |  |  |  |  |  |  |  |  |  |  |
| *Elphidium advenum* |  | 0.9 |  | 0.1 | 0.3 | 2.3 | 5.1 | 2.7 | 0.3 |  | 0.9 | 0.6 | 0.7 |  | 4.7 |  | 1.1 | 5.5 | 2.2 | 2.6 | 3.6 | 2.9 |  | 10.9 | 14.7 | 25.3 | 12.3 | 13.1 | 8.8 | 4.8 |
| *Elphidium clavatum* | 0.8 | 1.7 | 0.9 | 0.3 | 1.0 | 1.1 | 2.0 | 3.6 | 3.5 | 1.3 | 1.9 | 1.9 | 0.7 | 1.9 | 12.1 |  |  | 6.2 | 0.4 | 1.9 | 6.5 | 1.0 | 1.8 | 3.5 | 4.6 | 4.0 | 7.5 | 8.6 | 8.8 | 8.3 |
| *Elphidium somaense* | 0.6 | 2.8 | 1.2 | 0.9 | 1.0 | 4.5 |  |  | 0.6 | 2.0 | 2.8 |  |  |  | 0.7 | 3.0 | 17.9 | 2.1 | 5.2 | 20.7 | 26.8 |  |  | 2.0 | 2.8 | 2.8 |  |  |  |  |
| *Elphidium subarcticum* | 53.9 | 53.9 | 71.2 | 59.7 | 74.3 | 20.5 | 18.2 | 43.6 | 58.4 | 65.1 | 62.3 | 46.1 | 63.0 | 48.1 | 18.8 | 22.7 | 18.9 | 11.6 | 5.2 | 17.8 | 15.5 | 6.9 |  | 5.0 | 6.3 | 5.6 | 8.8 | 13.5 | 12.9 | 14.5 |
| *Elphidium subincertum* |  |  |  | 0.2 |  |  |  |  | 0.6 |  |  |  | 0.7 |  | 1.3 |  |  |  |  |  |  | 1.0 | 1.8 | 2.5 | 2.1 | 1.6 | 2.6 | 2.5 | 4.1 | 3.4 |
| *Fissurina laevigata* |  | 0.2 |  |  |  |  |  |  |  |  |  |  |  |  |  |  |  | 0.7 |  | 0.3 |  |  |  |  |  |  |  |  |  |  |
| *Melonis barleeanus* |  |  |  |  |  |  | 2.0 |  |  |  |  |  |  |  |  |  |  | 2.1 |  | 0.3 |  |  |  |  |  |  |  |  |  |  |
| *Nonionella globosa* |  |  |  |  |  |  |  |  |  |  |  |  |  |  | 0.7 |  |  |  |  |  |  |  |  |  |  |  |  |  |  |  |
| *Paracassidulina sagaminiensis* |  |  |  |  |  |  | 1.0 |  |  |  | 0.9 |  |  |  |  |  |  | 2.1 |  |  |  | 0.6 |  |  |  |  |  |  |  |  |
| *Pararotalia nipponica* |  |  |  |  |  | 1.1 |  |  |  |  |  |  |  |  |  |  |  | 0.7 |  |  |  |  |  |  |  |  |  |  |  |  |
| *Pseudononion japonicum* |  |  |  |  |  |  |  |  |  |  |  |  |  |  |  |  |  | 2.1 |  |  |  |  |  |  |  |  |  |  |  |  |
| *Pseudoparrella naraensis* |  | 0.9 |  |  |  | 2.3 |  |  | 0.3 |  |  |  | 0.6 |  |  |  |  | 2.7 |  |  |  |  |  |  |  |  |  |  |  |  |
| *Rosalina* sp. |  |  |  |  |  | 1.1 |  |  |  |  |  |  |  |  |  |  |  | 3.4 | 0.4 |  |  |  |  |  |  |  |  |  |  |  |
| **C.-P. Foram** | | | | | | | | | | | | | | | | | | | | | | | | | | | | | | |
| *Quinqueloculina seminula* | 0.6 |  | 0.3 |  |  | 2.3 | 3.0 | 2.7 | 4.2 | 2.6 |  |  | 0.7 |  | 8.7 |  |  | 2.1 |  |  | 4.2 | 0.6 |  |  | 1.5 | 2.9 | 8.4 | 3.1 | 1.6 | 1.0 | 6.2 |
| *Sigmoilopsis* sp. |  |  |  |  |  |  |  |  | 0.3 |  |  |  |  |  |  |  |  | 2.1 |  |  |  |  |  |  |  |  |  | 2.2 |  | 0.7 |
| Total No. of Benthic Foram | 514 | 531 | 347 | 1341 | 575 | 88 | 99 | 110 | 310 | 152 | 212 | 362 | 138 | 52 | 149 | 66 | 95 | 146 | 269 | 309 | 168 | 102 | 57 | 201 | 715 | 249 | 228 | 244 | 194 | 145 |
| Species Number S | 9 | 12 | 9 | 10 | 11 | 15 | 10 | 8 | 13 | 8 | 9 | 7 | 8 | 6 | 13 | 8 | 9 | 25 | 9 | 13 | 11 | 7 | 4 | 8 | 11 | 9 | 8 | 8 | 7 | 10 |
| Percentage of A. Foram | 27.8 | 24.1 | 8.9 | 26.6 | 8.7 | 23.9 | 21.2 | 12.7 | 2.6 | 11.2 | 18.9 | 1.1 | 7.2 | 13.5 | 8.7 | 53.0 | 24.2 | 7.5 | 10.4 | 13.6 | 3.0 | 1.0 |  | 0.1 |  |  |  | 0.4 |  | 2.1 |
| Percentage of C.-H. Foram | 71.6 | 75.9 | 91 | 73.4 | 91.3 | 73.9 | 75.8 | 84.5 | 92.9 | 86.2 | 81.1 | 98.9 | 92.0 | 86.5 | 82.6 | 47.0 | 75.8 | 88.4 | 89.6 | 82.2 | 96.4 | 99.0 | 100 | 98.5 | 96.9 | 91.6 | 94.7 | 98.0 | 99.0 | 91.0 |
| Percentage of C.-P. Foram | 0.6 |  | 0.3 |  |  | 2.3 | 3.0 | 2.7 | 4.5 | 2.6 |  |  | 0.7 |  | 8.7 |  |  | 4.1 |  | 4.2 | 0.6 |  |  | 1.5 | 2.9 | 8.4 | 5.3 | 1.6 | 1.0 | 6.9 |
| Total No. of Planktic Foram |  | 4 | 1 | 2 |  | 1 |  |  |  |  | 2 | 1 |  |  |  |  | 1 | 9 | 1 |  | 1 |  |  | 1 | 8 |  |  |  |  |  |
| Species Diversity H(s) | 1.36 | 1.48 | 1.02 | 1.28 | 1.02 | 2.02 | 1.80 | 1.50 | 1.36 | 1.23 | 1.22 | 1.13 | 1.14 | 1.35 | 1.98 | 1.57 | 1.75 | 2.67 | 1.31 | 1.95 | 1.81 | 1.15 | 0.54 | 1.17 | 1.48 | 1.60 | 1.57 | 1.61 | 1.59 | 1.66 |
| Equitability E | 0.62 | 0.60 | 0.46 | 0.56 | 0.42 | 0.75 | 0.78 | 0.72 | 0.53 | 0.59 | 0.56 | 0.58 | 0.55 | 0.75 | 0.77 | 0.75 | 0.80 | 0.83 | 0.60 | 0.76 | 0.75 | 0.59 | 0.39 | 0.56 | 0.62 | 0.73 | 0.76 | 0.77 | 0.82 | 0.72 |
| Total No. of B. F. in 20ml of sd. | 1028 | 1062 | 694 | 2682 | 1150 | 88 | 99 | 110 | 310 | 304 | 424 | 362 | 138 | 52 | 149 | 66 | 95 | 292 | 538 | 309 | 168 | 102 | 57 | 804 | 1430 | 996 | 912 | 488 | 388 | 1160 |
| Total No. of P. F. in 20ml of sd. |  | 8 | 2 | 4 |  | 1 |  |  |  |  | 4 | 1 |  |  |  |  | 1 | 18 | 2 |  | 1 |  |  | 4 | 16 |  |  |  |  |  |

A.=Agglutinated, C.-H.= Calcareous-Hyaline, C.-P.=Calcareous-Porcelaneous, B. F.=Benthic Foraminifera, P. F.=Planktic Foraminifera, sd.=sediments