# Peer review of "Historical record of the effects of anthropogenic pollution on benthic foraminifera over the last 110 years in Gamak Bay, South Korea"

_Biogeosciences, 2017_

## Referee Comment (RC1) · Anonymous Referee #1 · 24 Nov 2017

The authors discuss anthropogenically induced environmental changes over the last 100 years in South Korean bay by traditional method. Benthic foraminifera are one of the most useful organisms for environmental assessment, because they can be used not only as recent environmental indicator but also as paleoenvironmental indicator. The manuscript is generally well written, but I think some discussions should be added before its acceptance.

The most critical point is the age model of the cores. The unusual distributions of 210Pb in the sediment cores should be evaluated carefully. It may indicate sediment mixing or erosion. Indeed, the authors indicate the possibility of the effect of the dredging on

benthic foraminifera. Discussion with raw data plots of 210Pb is needed. Accumulation rates of geochemical elements and benthic foraminiferal number (BFAR) are more suitable because sedimentation rates vary over time.

Diagenetic processes influence TOC, TN, and C/S. This needs to be evaluated. Natural variability, such as precipitation and flood, affect the benthic foraminiferal assemblage. This needs to be discussed.

The authors discuss the effects of pollution on benthic foraminifera mainly based on the changes in the relative abundance of Elphidium subarcticum, the bioindicator of organic pollution in Gamak Bay. Ammonia beccarii, which is known as the pollution tolerant species in coastal seas throughout the world, decreased after the progress of pollution. In addition, Buccella frigida has also been reported as tolerant species of contaminated waters. Discussion of this manuscript is mainly based on own data, and many references on foraminifera as indicators of human impacts are missing. This point makes this manuscript local subject.

The authors correlate the benthic foraminiferal assemblages among sites, but the validity is not discussed. Why are those assemblages comparable? If you compare the assemblages, you should perform cluster analysis for all samples. The authors correlate the Ab-Bf-Ea assemblage of site 11285 with the Ab-Ea-Ec assemblage of site 10863, but the authors point out that the Ab-Bf-Ea assemblage has already been affected by anthropogenic impact. If this is true, these two assemblages are not comparable. Moreover, the authors correlate the Es, Ab-Es-Th and Es assemblages of site 11285 with the Es-Ab assemblage of site 10863. The authors argue that the Es, Ab-Es-Th and Es assemblages of site 11285 were affected by local mussel farming, and Es-Ab assemblage of site 10863 was affected by local oyster farming. Mussel farming and oyster farming are different pollutant sources, and these two pollutant sources are not correlated. So, the changes of foraminiferal assemblages in two sites are distinctive.

The authors argue that pollution has already started in the inner part based on

foraminiferal diversity and abundance. However, it may indicate natural environmental gradient (i.e. salinity gradient). Foraminiferal diversity is usually low in the brackish area because of its low salinity environment. So, please indicate spatial distribution of surface and bottom water salinity in Gamak Bay.

p. 12 line 20 to 26: The authors argue that the rapid increase in the abundance of Ammonia beccarii have been caused by an improvement in the benthic environment via dredging. However, geochemical data do not show the improvement in the benthic environment.

5.4 Pollution variation: Many geochemical elements negatively or positively correlated with foraminiferal species, but the authors focus on the TOC content. How can you separate the TOC content with other types of pollutants?

Figure 9 shows exactly the same thing as figure 6. I think figure 9 is not necessary.

minor suggestions

P. 1 Line 31: E. subarcticum should be italic.

P. 2 Line 2: Mytilus galloprovincialis should be italic.

P. 10 Line 18: A. beccarii should be italic.

Best regards,

---

## Referee Comment (RC2) · Anonymous Referee #2 · 27 Nov 2017

The manuscript is on an interesting topic but is very descriptive and lacks implications for a wider audience.

The abstract is much too long and full of details; it would instead need a synthetic treamtent of problem, the approach, the most significant results and some conclusive statements.

Although the introduction is solid in terms of describing the problem (pollution) and the approach to reconstruct its history (forams and geochemical data from sediment cores), there is no interesting hypothesis raised and the approach is therefore very descriptive.

[Figure]

210Pb is used for dating and to estimate sedimentation rates, it is not a geochemical signal

The second half of the text under the heading "Study area", starting with "The bay is home to. . ." would at least partly better fit into the introduction. It would offer the chance to formulate more interesting study aims / hypotheses.

Provide equation for diversity indices and make clear which of the two (H' or J) you used were in the text. What is the justification for using exacly these 2 indices?

Figs 6, 7, 8 are only presented in the discussion. They all deal with results and should be moved (together wit the text that relates to them) to the Result section.

Fig 9 is presented only in the conclusion. This is totally inappropriate. Please use this figure for a thorough discussion. I am not sure, however, if this figure is needed at all.

The conclusions are actually a summary and real conclusions from this study are lacking.

---

## Author Comment (AC1) · 4 Dec 2017

1. The age model is very important in this manuscript. The data of Pb-210 acquired from core samples should be evaluated carefully, because of sediment mixing or erosion. Authors were observed carefully to the cross section of core sediment (sediment structure, gain size composition variation of sediment). Sediments composed of homogeneous fine-grained mud facies with 30.02–33.25% silt and 66.10–68.84% clay accumulated in three cores. We could not find a distinct sedimentary structure. Excess 210Pb, however, did not decline exponentially downward in the cores, and the highest 210Pb values of cores 10863 and 11285 were found at depths of 3 cm and 7 cm,

respectively. This abnormality beneath the surface sediment has also been found in other coastal environments (e.g., Ruiz-Fernández et al. 2003; Lubis 2006), probably due to sediment reworking such as waves, current, or bioturbation on the top layer. From the excess 210Pb activity profile shown in Table 4, we used the Constant Rate of Supply (CRS) model (Appleby and Oldfield, 1992) for calculating the sediment age and accumulation rates, assuming a constant rate of supply of excess 210Pb per unit time. Yes, of course, I know about the relationship between accumulation rates of geo-chemical elements and benthic foraminiferal number (BFAR) (Tsujimoto et al., 2008). It may be also useful tool to know about them. . 2. Tsujimoto et al. (2008) reported that the steep trend in both the C/S and C/N after the 1990's in Osaka Bay may be due not to the environmental change but instead due to the progress of sulfate reduction by bacteria at a very early stage of diagenesis (Sampei et al., 1997). In our data, the steep variation appeared at TOC of 21 cm (1978 yr) in core 10863 (Figure 2-J). I think that it is too young to occur diagenesis. 3. I know well about A. beccarii that is known as the pollution tolerant species. The genus, Ammonia is usually described as being tolerant of all kinds of stress conditions, including organic and heavy metal pollution (Armynot du Châtelet, et al., 2004; Ferraro et al., 2006; Frontalini and Coccioni, 2008). Yasuhara et al. (2012) reported that the tolerant genus Ammonia increases and the sensitive genus Elphidium decrease with increasing eutrophication and resulting hy-poxia/anoxia at various locations, including Osaka Bay (Tsujimoto et al., 2006, 2008), Gulf of Xexico (Sen Gupta et al., 1996; Sen Gupta and Platon 2006; Rabalais et al., 2007), Chesapeake Bay (Karlsen et al., 2000), San Francisco Bay (McGann 2008), Long Island Sound (Thomass et al., 2000), and the Bay of Biscay (Irabien et al., 2008). In this study, abundance frequency of A. beccarii and E. subarcticum was varied re-markably (Figure 3-III-D, I). A. beccarii decreased and E. subarcticum increased with increasing eutrophication and hypoxia in assemblages of the northwestern area. Espe-cially, variation of foraminiferal assemblage from Es through Ab-Es-Th to Es distributed between 1988 and 2014 indicates from worsening through recovering to worsening again in benthic ecology and the sedimentary environment. A. beccarii may be rapidly

decreased with rapid deterioration of habitat environment, although A. beccarii is a pollution tolerant species. E. subarcticum, however, was correlated distinctly with variation of benthic ecology and the sedimentary environment. Therefore, the authors focused on the E. subarcticum. 4. I tried to compare to benthic foraminiferal assemblage in equal time zone, and understand the difference of habitat conditions. 5. Inflow into northwestern area of freshwater through very small stream little affected to the salinity environment, because of extremely small quantity. If brackish area was distributed extensively, mussel farm could not developed. 6. Yes, geochemical data do not show the improvement in the benthic environment after dredging. It may be caused by thick polluted sediment body ("pollution storage"). 7. The main factor of pollution in this study is organic materials formed by mussel farm. The TOC is representative geochemical element to organic material. Therefore, the authors focused on the TOC content. 8. I don't think so. Figure 9 suggest that Figure 6 was caused by sea water movement. 9. Thank you for your kindness. I will revise.